# TuneAhead: Predicting Fine-tuning Performance Before Full Training Begins

**Yuxiang Luo**[1] **Haonan Long**[1] **Chen Wang**[1] **Qiqi Duan**[1] **Xiaotian Lin**[1] **Yanwei Xu**[2] **Yuyu Luo**[1]
**Weikai Yang**[1] **Nan Tang**[1]

## Abstract

Fine-tuning large language models (LLMs) is compute-intensive and error-prone: model performance depends sensitively on data quality and hyperparameter choices, and naïve runs can even degrade model performance. This raises a practical question: *can we predict fine-tuning performance before committing to a full training run?* We present TUNEAHEAD, a lightweight framework for pre-hoc prediction of fine-tuning performance. TUNEAHEAD encodes each candidate run as a meta-feature vector that combines static dataset descriptors with dynamic probe features from a short standardized probe. A predictor maps these features to performance estimates, while SHAP-based attributions provide interpretable diagnostics that reveal which specific features drive the prediction. Across 1,300+ fine-tuning runs on Qwen2.5-7B-Instruct, TUNEAHEAD consistently outperforms strong baselines such as Early-Stop Extrapolation and ProxyLM. On a held-out test set of 370 runs, TUNEAHEAD achieves an RMSE of 1.47 percentage points and places 95.1% of predictions within $\pm 3$ percentage points of the true score. These accurate continuous predictions support practical go/no-go screening policies that can reduce unnecessary full fine-tuning while retaining most promising runs.

## 1. Introduction

Fine-tuning large language models (LLMs) has become the standard path to domain adaptation, but it remains costly and unpredictable: performance depends sensitively on data quality and hyperparameter choices, and naïve runs can even *degrade* downstream performance in real-world pipelines (Barnett et al., 2024). For practitioners, the key

question is not only ***how*** to fine-tune, but increasingly ***whether*** a run is worth doing at all.

**Predicting fine-tuning success.** Consider a healthcare provider deciding whether to fine-tune a general LLM on a clinical dataset: a failed run may consume hundreds of GPU hours yet underperform the base model. Without prediction, practitioners often discover *only after training* that performance falls short, wasting both time and budget (Figure 1(A)). This raises a crucial question: *can we predict fine-tuning performance before committing to the full fine-tuning run?*

**Prior art and their limitations.** Scaling-law analyses (Kaplan et al., 2020) capture general trends across models and datasets but offer limited insight for a *specific* dataset. Proxy models such as COSMOS (Wang et al., 2025) and short-horizon extrapolations (Kuramoto & Suzuki, 2025) demonstrate that low-cost prediction is feasible. However, they aggregate all features into a single score, conflating the base model's own characteristics and limitations with dataset properties, leaving practitioners unable to answer the crucial question of 'why' a run might fail and thus unable to make targeted improvements to avoid costly failures.

**Predicting with TUNEAHEAD.** We introduce TUNEAHEAD, a diagnostic prediction framework that predicts fine-tuning performance *before* the full fine-tuning run is launched. The core idea is to capture two complementary categories of low-cost features. The first is *static dataset descriptors*, which are computed from the dataset itself to provide a foundational, model-agnostic assessment of its intrinsic quality (*e.g.,* lexical diversity, data size). The second category is *dynamic probe features,* which are extracted from a short probe run (*e.g.,* early loss decay, gradient stability); their unique advantage is capturing the model-specific learnability of the data, revealing early signs of optimization instability or data-model mismatch that are invisible to static analysis alone. A lightweight gradient-boosting predictor maps these features to expected performance, while SHAP-based attribution (Lundberg & Lee, 2017) converts predictions into explanations that reveal which dataset properties matter most. We define a run as 'success' if its predicted score exceeds a predefined practical threshold, and 'failure' otherwise.

---

[1]The Hong Kong University of Science and Technology (Guangzhou) [2]Huawei Technologies Ltd.. Correspondence to: Nan Tang <nantang@hkust-gz.edu.cn>.

*Proceedings of the 43rd International Conference on Machine Learning*, Seoul, South Korea. PMLR 306, 2026. Copyright 2026 by the author(s).

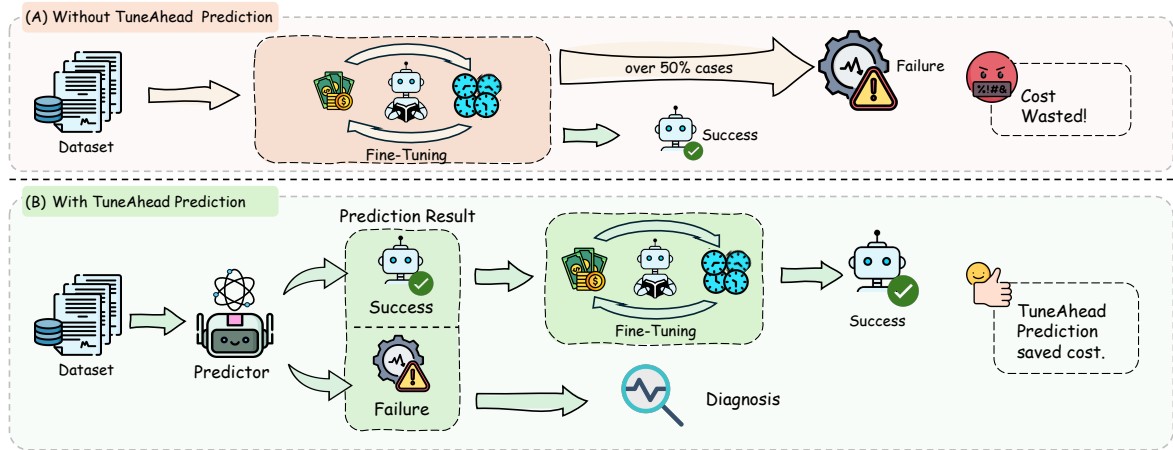

*Figure 1.* Predicting fine-tuning performance: **(A)** Without **TUNEAHEAD**: failed runs are only identified after training, wasting computational resources and time. **(B)** With **TUNEAHEAD**: low-cost features predict performance in advance, enabling go/no-go decisions and diagnosis for the failure cases.

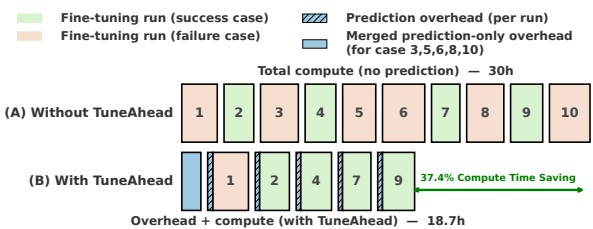

*Figure 2.* Compute time for 10 runs without **TUNEAHEAD** (A) vs. with **TUNEAHEAD** (B).

With **TUNEAHEAD** (see Figure 1(B)), practitioners can detect unpromising runs *before* training, cutting substantial compute costs while gaining actionable guidance for data refinement. By preventing wasted resources, **TUNEAHEAD** overcomes key limitations of prior work, offering a nearly training-free, interpretable, and dataset-aware approach to reliable fine-tuning.

Figure 2(A) shows that without prediction, all 10 runs (including failures) require ∼30 GPU-hours. With **TUNEAHEAD** (Figure 2(B)), failures are flagged in advance (hatched/blue), so only promising runs are trained, reducing compute to ∼18.7 GPU-hours–a 37.4% saving in this illustrative 4/6 success/failure candidate pool. The realized saving can vary with the candidate-pool composition and the chosen success threshold, and we provide a threshold-sensitive analysis in Section 5.

**Contributions.** We make the following contributions.

(1) *The problem of predicting fine-tuning performance.* We cast fine-tuning outcome prediction as a *pre-hoc, diagnostic meta-learning task*. This formulation supports early go/no-go decisions and principled dataset ranking before expensive training is attempted. (Sec. 3)

(2) *The* **TUNEAHEAD** *framework.* We design a hybrid feature space that combines static dataset descriptors with dynamic probe features, and pairs it with a lightweight predictor and SHAP-based attributions, yielding both accurate predictions and interpretable diagnostics. (Sec. 4)

(3) *Extensive experiments.* We conduct over 1,300 fine-tuning runs on Qwen2.5-7B-Instruct (Qwen Team, 2025; Yang et al., 2025a), evaluated on several benchmarks and different types of tasks. **TUNEAHEAD** consistently outperforms strong baselines such as Early-Stop Extrapolation, ProxyLM, and Early-Dynamics baseline. (Sec. 5)

**Conflict of Interest Disclosure.** Yanwei Xu is affiliated with Huawei Technologies Ltd. The remaining authors are affiliated with The Hong Kong University of Science and Technology (Guangzhou). The authors declare no financial conflicts of interest that directly affect the design, evaluation, or conclusions of this work.

## 2. Related Work

Our work sits at the junction of two threads: *LLM performance prediction* and *dataset-level quality assessment*, with connections to fine-tuning dynamics and interpretability. We position **TUNEAHEAD** as a pre-hoc, dataset-level approach that is both *predictive* and *diagnostic*.

**LLM performance prediction.** Early efforts predict performances by extrapolating short training curves (Domhan et al., 2015), an approach that struggles with modern LLM fine-tuning where dynamics can be non-monotonic and late-emerging. Recent works (*e.g.,* LENSLLM (Zeng et al., 2025), (Lin et al., 2024)) analyze during-training behavior (such as the pre-power phase) using NTK (Jacot et al., 2020) and scaling laws (Kaplan et al., 2020) on small subsets, sharing that the philosophy of 'fitting small to predict large' involves extrapolating the effect of data quantity.

However, they do not diagnose the specific root causes of performance degradation (*e.g.,* data repetition or gradient instability) and profile data quality. A parallel line uses inexpensive surrogates: proxy heads or smaller models can predict final accuracy at a low cost (COSMOS (Wang et al., 2025), PROXYLM (Anugraha et al., 2024), probing-based predictors (Zhu et al., 2022)). However, these methods typically return a single, entangled score that mixes model bias with data properties, offering limited visibility into *why* a run succeeds or fails. Our formulation departs from prior work by combining pre-hoc prediction with an explicit, dataset-level diagnosis.

**Data quality assessment.** A complementary literature scores data quality at the *instance* level—*e.g.,* Data Cartography (Swayamdipta et al., 2020) and Data Shapley (Ghorbani & Zou, 2019)—and more recently curates instruction-tuning data via refinement pipelines (Refine-n-Judge (Cayir et al., 2025)). Others move toward holistic descriptors: Dataset Nutrition Labels (Holland et al., 2018), distributional measures like MAUVE (Pillutla et al., 2021), and generative teaching evaluations (GENTLE (Aoyama et al., 2023)). They improve transparency but generally stop short of *predicting* a dataset's fine-tuning payoff. **TUNEAHEAD** closes this gap by treating each dataset as a meta-instance and tying aggregated *static* descriptors and *early* interaction features to downstream performance.

**Fine-tuning dynamics and interpretability.** Work on early training signals (*e.g.,* gradient/loss dynamics) informs our choice of low-cost probes (Jastrzebski et al., 2020; Hao et al., 2019). For interpretability, we adopt model-agnostic SHAP attributions (Lundberg & Lee, 2017) to move beyond predicting toward actionable diagnosis at the dataset level.

## 3. Problem of Predicting Fine-tuning Performance

This section formalizes the task of predicting fine-tuning performance before actual training. Given a dataset–hyperparameter pair, we seek a **low-cost** predictor that approximates the **expensive ground-truth** score that would be obtained after completing full fine-tuning and evaluation. Specifically, let $M$ be a base LLM (*e.g.,* Qwen2.5-7B-Instruct), and $A$ denote a general fine-tuning algorithm. Fine-tuning model $M$ on the dataset–hyperparameter pair $(D_i, H_j)$ produces an adapted model:

$$M'_{i,j} = A(M, D_i, H_j).$$

We then evaluate $M'_{i,j}$ on a downstream benchmark $T$ (*e.g.,* MMLU) to obtain the *ground-truth performance score* $R_{i,j}$. However, acquiring this score is expensive as it requires a full fine-tuning and evaluation cycle, which motivates the need for prediction.

We thus seek a *low-cost prediction function F* that consumes a *meta-feature vector* $V_{i,j}$ describing the dataset and hyperparameter configuration, producing a predicted performance score $P_{i,j}$ that well approximates the ground-truth score:

$$P_{i,j} = F(V_{i,j}) \quad \text{with} \quad P_{i,j} \approx R_{i,j}.$$

The primary output of $F$ is continuous. In deployment, a practitioner may optionally choose a task-specific threshold $\tau$ and convert the prediction into a go/no-go decision, *e.g.,* launching full fine-tuning only when $P_{i,j} \geq \tau$. This thresholding step is a deployment policy applied after prediction; it does not affect the regression model or the continuous evaluation metrics.

The prediction function $F$ is trained on a meta-dataset of past fine-tuning experiments drawn from an empirical distribution Dist over pairs $(D_i, H_j)$. Minimizing an appropriate loss $\Delta$ (such as mean squared error), we solve

$$\min_{F} \ \mathbb{E}_{(D_i, H_j) \sim \text{Dist}} \Big[ \Delta\big(F(V_{i,j}), R_{i,j}\big) \Big].$$

This formulation supports several key practical goals: (i) *making go/no-go decisions* before expensive fine-tuning; (ii) *ranking dataset and hyperparameter settings* for resource allocation; and (iii) *diagnosing fine-tuning performance* by linking predictions to dataset and hyperparameter characteristics.

**Why pre-hoc failure prediction is feasible.** Failed fine-tuning runs often leave clear, low-cost features. Examples include dataset-model mismatch (*e.g.,* high reference perplexity), redundancy or limited diversity (flat or noisy short-horizon progress), and unstable optimization dynamics (volatile gradients and irregular loss decay). Notably, failures are often easier to detect than successes: even a single **strong deficiency can reliably indicate likely failure**, enabling early *rule-out* and data-centric remedies at minimal cost.

## 4. The TUNEAHEAD Framework

### 4.1. Design Goals and Framework Overview

**Design Goals.** We identify three key goals that guide our framework design for predicting fine-tuning performance: *(G1) Low-cost yet informative features:* The meta-features must respect a computational cost budget while still effectively making predictions. *(G2) Reliable and generalizable prediction:* Predictions must be accurate, well-calibrated, and generalizable across diverse datasets. *(G3) Diagnostic interpretability:* Predictions must come with human-interpretable attributions that highlight actionable guidance for targeted improvement.

**Framework Overview.** To satisfy these three goals, TUNEAHEAD is structured into two complementary stages: meta-dataset curation (stage 1) and predictive & diagnostic modeling (stage 2). Figure 3 illustrates the framework overview. Stage 1 constructs a compact meta-feature vector $V_{i,j}$ for each dataset-hyperparameter pair by combining static features that summarize dataset-intrinsic properties and dynamic features (G1) that capture early training behavior via a short, fixed-budget probe run. These features are fed into stage 2 for predicting model performance $P_{i,j}$ (G2) along with detailed diagnostic attributions (G3). These explanations identify specific failure modes such as data mismatch, redundancy, or instability, helping enable early rejection of failure runs and guiding focused data-centric improvements.

### 4.2. Stage 1: Meta-Dataset Curation

The predictive capability of **TUNEAHEAD** heavily depends on how well the meta-feature vector $V_{i,j}$ characterizes each $(D_i, H_j)$. To balance informativeness with efficiency, $V_{i,j}$ integrates two complementary categories of features: *static features* derived from the dataset itself, providing a model-agnostic prior on dataset quality, and *dynamic features* probed from the base model $M$ via a short, fixed-budget run, exposing early signs of instability or mismatch. To ensure our meta-feature vector $V_{i,j}$ is both predictive (effective) and computationally lightweight (efficient), we construct it through a rigorous pipeline.

**Feature Selection Process.** Instead of using an arbitrary feature set, we initiated our framework with a broad candidate pool of over 50 features. We implemented a SHAP-guided selection pipeline to distill this pool into a compact and interpretable feature set; the full procedure is detailed in Appendix A.3. To ensure a clean evaluation protocol, all feature-selection decisions are made before touching the held-out test split. In particular, the preliminary Light-GBM model, SHAP computation, percentile-based pruning, directional-consistency filtering, and redundancy checks are performed only on the training/validation portion of the meta-dataset. Once the final set of features is fixed, the held-out test split is used only for final evaluation.

Specifically, we trained a preliminary LightGBM regressor on the candidate set and filtered features based on three criteria:

**Global Importance ($s_f$).** For each feature $f$, we computed the mean absolute SHAP value $s_f = \frac{1}{N} \sum_{i=1}^{N} |\phi_{i,f}|$, where $\phi_{i,f}$ is the SHAP value of feature $f$ for run $i$, and $N$ is the number of runs. Features with $s_f$ below the 15th percentile were pruned as non-informative.

**Directional Consistency ($c_f$).** We defined consistency as $c_f = \text{sign}(\rho_f) \cdot \rho_f$, where $\rho_f$ is the Spearman correlation between feature $f$ and the target. We removed features with unstable directions ($c_f < 0.2$) to ensure interpretability (*e.g.,* "lower loss should generally indicate easier optimization").

**Redundancy Pruning.** We iteratively removed highly correlated features unless their exclusion significantly degraded the cross-validated RMSE (i.e., $\Delta\text{RMSE} > 0.01$).

This data-driven process reduced the pool from over 50 features to the 24 most discriminative features (14 static, 10 dynamic). The final set of *24* meta-features is chosen to be both *predictive* and *lightweight*, so that the vector $\mathbf{V}_{i,j}$ jointly encodes (i) dataset quality before full fine-tuning and (ii) model learnability during the short probe. Below we summarize each group with brief computation notes; full definitions remain in Appendix B.

**Static Features.** These are dataset-intrinsic descriptors computed prior to training. We retained 14 features across four categories:

**A. Global statistics.** Basic aggregations from a single tokenization pass provide powerful signals. *Dataset Size* bounds "how much there is to learn"; *Token Lengths* (mean $\mu_{\text{len}}$ and std $\sigma_{\text{len}}$) warn of memory pressure; the *Input–Output Length Ratio* separates compressive tasks from elaborative ones, directly influencing adaptation speed; and the *Special-Character Ratio* flags insufficiently cleaned text.

**B. Lexical diversity.** To distinguish simple lexical repetition from noise caused by excessively rare or idiosyncratic tokens, we compute the Type–Token Ratio and *N-gram Repetition*. We also estimate *Instruction Complexity* via lightweight parse-tree depth to detect prompts that under-challenge the model.

**C. Semantic diversity.** Surface statistics miss semantic redundancy, so we add *Approximate Duplicates* (using Min-Hash to avoid quadratic checks) and *Embedding Outlier Ratio* (applying a robust 3-sigma rule: $\frac{\left|\{E(s):\|E(s)-\mu\|>3\sigma\}\right|}{N}$, where $E(s)$ is the embedding of sample $s$, $\mu$ and $\sigma$ are the mean and standard deviation of embeddings over the dataset, and $N$ is the number of samples). We also measure *IO-Semantic Similarity* and *Output Semantic Diversity* to identify uninformative paraphrasing or incoherent scatter.

**D. Model-based complexity.** We include difficulty measured by computing *Reference Perplexity*. The mean tracks domain mismatch, while the standard deviation reveals within-dataset heterogeneity. We also compute *KL-divergence* between dataset and pretraining-corpus token distributions to quantify vocabulary shift, and *Answer Groundedness* to detect over-copying vs. hallucination.

**Dynamic Features.** While static features describe the dataset in isolation, they cannot capture how a specific base model interacts with the dataset under a particular fine-

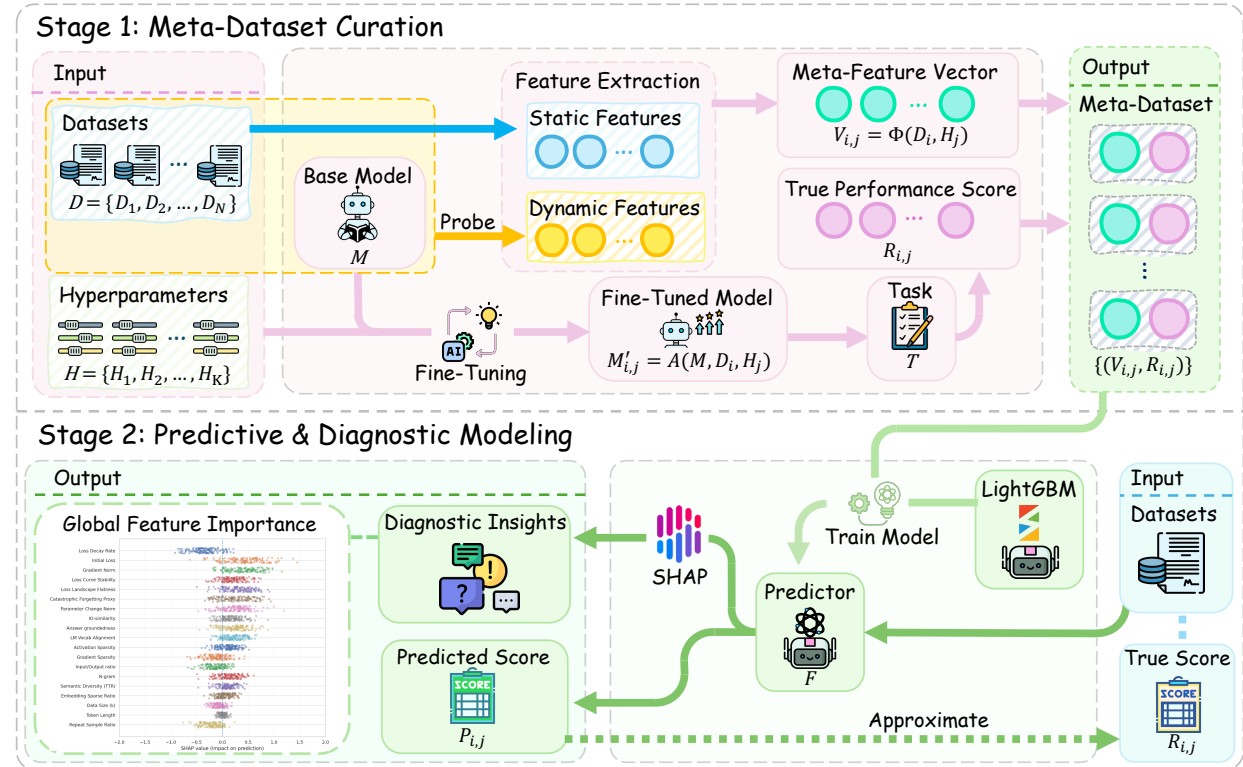

*Figure 3.* **TUNEAHEAD** Overview. Stage 1 (Meta-dataset curation) builds meta-feature vectors $V_{i,j}$ by combining static features with dynamic features. Stage 2 (Predictive & Diagnostic Modeling) maps $V_{i,j}$ to performance predictions and uses SHAP for diagnostics.

tuning configuration. We therefore run a standardized 100-step probe for each dataset–hyperparameter pair $(D_i, H_j)$. The resulting dynamic features should be interpreted as *interaction features*: they reflect not only the dataset, but also how the chosen hyperparameters affect early optimization behavior.

**E. Loss dynamics.** We capture initial difficulty via *Initial Loss*. Crucially, we compute *Loss Decay* by fitting a robust linear slope to the log-loss curve; more negative slopes indicate faster adaptation. *Loss Stability* is quantified by the variance of the probe losses: $\sigma_L^2 = \frac{1}{T} \sum_{t=1}^{T} (L_t - \bar{L})^2$, where $L_t$ is the training loss at probe step $t$ and $\bar{L}$ is the mean loss. Larger values of $\sigma_L^2$ indicate higher loss volatility, which signals noisy optimization.

**F. Gradient signals.** The mean and variance of the *Gradient Norm* indicate explosion or vanishing risks. *Gradient Consistency* reflects update coherence, while *Gradient Sparsity* flags under-utilized capacity.

**G. Generalization cues.** We monitor the *Parameter Change Norm*—too little movement fails to learn, while too much often correlates with overfitting. Finally, we compute a *Landscape Flatness* proxy by perturbing parameters and observing loss change; flatter regions in the probe tend to generalize better.

### 4.3. Stage 2: Prediction and Diagnostic Model

Based on the meta-feature vector extracted in stage 1, we aim to train a lightweight predictor that both well approximates the fine-tuning performance $R_{i,j}$ (G2) while also explaining why a run is likely to succeed or fail (G3). We adopt LightGBM, a gradient-boosted tree model particularly well-suited for heterogeneous, tabular meta-features. As demonstrated in Appendix D, LightGBM achieves accuracy comparable to state-of-the-art alternatives (*e.g.,* SVR) while providing significantly better interpretability and scalability. These properties make it a principled design choice rather than a simple off-the-shelf baseline.

In addition to its powerful prediction capability, LightGBM seamlessly integrates TreeSHAP, a theoretically rigorous method for Shapley value attribution. The SHAP framework decomposes each prediction $P_{i,j}$ into additive contributions from individual meta-features. Unlike black-box or proxy baselines that provide only an opaque overall score, our model delivers transparent attributions. For example, a predicted failure can now be traced back to low lexical diversity (static feature) or unstable gradient norms (dynamic probe feature), which guide targeted improvement.

# 5. Experiments

**Setup.** We validate **TUNEAHEAD** across diverse data and hyperparameter settings on the MMLU task, using Qwen2.5-7B-Instruct, Llama-3-8B-Instruct, and Qwen2-0.5B as base models. MMLU is chosen as a representative multi-domain knowledge benchmark to test whether fine-tuning outcomes are predictable under heterogeneous subject distributions and hyperparameter variations. Each meta-example corresponds to a dataset–hyperparameter pair $(D_i, H_j)$ drawn from a predefined LoRA search space, rather than from a separately optimized configuration for each dataset. Thus, hyperparameters are treated as part of the candidate-run input to be predicted, not as a hidden tuning procedure. On the predictor side, we use the same feature pipeline and a fixed LightGBM configuration across experiments, including learning rate 0.05 and number of leaves 4. All implementation details, data curation protocols, model training procedures, calibration experiments, and baseline implementations are provided in Appendix A. Ground-truth labels are seed-averaged MMLU test accuracy from full LoRA fine-tuning; unless stated otherwise, we average over three seeds and evaluate on a held-out test split. To test generalization beyond multiple-choice knowledge benchmarks and assess robustness across task formats, we further conduct experiments on **TruthfulQA** (Lin et al., 2022), which evaluates instruction fidelity and factual consistency, and on an open-ended summarization task (**SAMSum**), which involves long-form generative outputs.

**Baselines.** We compare **TUNEAHEAD** against (1) literature/practice-inspired baselines and (2) ablation variants: (i) *Early-Stop Extrapolation* — linear extrapolation from the 100-step probe validation loss (Domhan et al., 2015; Adriaensen et al., 2023); (ii) *Early Training Dynamics* — a method hypothesizes that the speed of loss decay in the initial steps is the strongest predictor of final convergence. We implemented this established heuristic using the slope of the probe loss curve. (Paul et al., 2023; Zhou et al., 2021); (iii) *Domain-Proxy Baseline* — method using the pretrained model's perplexity on the target data as a standard proxy for "difficulty" or "distribution shift" to do data selection or prediction without needing full fine-tuning (Gururangan et al., 2020; Harada et al., 2025; Huang et al., 2022); (iv) *ProxyLM* (Anugraha et al., 2024) — regress on proxy models' accuracy (*e.g.,* small LMs) optionally combined with dataset features (*e.g.,* TTR, vocabulary size and average token length); (v) **TUNEAHEAD**-*Static-Only* — LightGBM trained on static dataset-intrinsic features only; (vi) **TUNEAHEAD**-*Dynamic-Only* — LightGBM trained on dynamic 100-step probe features only.

**Metrics.** We report complementary metrics on the test set, all in percentage-point (pp) units for accuracy predictions: *RMSE* quantifies prediction error; $R^2$ quantifies explained

*Table 1.* **Main results on predicting fine-tuning performances (MMLU).** Arrows indicate direction: RMSE↓, $R^2$ ↑, $r$ ↑, Acc@$k$pp↑. **TUNEAHEAD** Pareto-dominates all baselines.

| Method | RMSE ↓ | $R^2$ ↑ | $r$ ↑ | Accuracy (±$k$pp) ↑ | | |
|---|---|---|---|---|---|---|
| | | | | $k{=}1$ | $k{=}2$ | $k{=}3$ |
| Early-Stop Extrapolation | 7.43 | 0.81 | 0.90 | 11.2 | 23.9 | 32.8 |
| Domain-Proxy Baseline | 6.58 | 0.85 | 0.92 | 8.6 | 22.0 | 32.8 |
| **TUNEAHEAD**-Static-Only | 3.50 | 0.96 | 0.99 | 14.9 | 32.8 | 49.3 |
| **TUNEAHEAD**-Dynamic-Only | 3.38 | 0.96 | 0.99 | 19.8 | 35.8 | 55.6 |
| Early-Dynamics Baseline | 3.33 | 0.96 | 0.98 | 29.9 | 50.0 | 67.5 |
| ProxyLM | 2.11 | 0.98 | 0.99 | 40.7 | 67.9 | 85.8 |
| **TUNEAHEAD (Full)** | **1.47** | **0.99** | **0.99** | **50.0** | **82.5** | **95.1** |

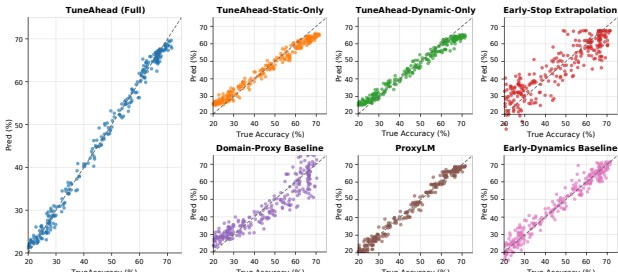

*Figure 4.* **Predicted vs True accuracy across methods.** The diagonal line ($y{=}x$) indicates a perfect prediction.

*Table 2.* Cross-model generalization with baselines. Arrows indicate direction: RMSE↓, $R^2$ ↑, $r$ ↑, Acc@$k$pp↑.

| Base Model | Method | RMSE↓ | $R^2$ ↑ | $r$ ↑ | Accuracy (±$k$pp) ↑ | | |
|---|---|---|---|---|---|---|---|
| | | | | | $k{=}1$ | $k{=}2$ | $k{=}3$ |
| Llama-3-8B-Instruct | Domain-Proxy Baseline | 10.01 | 0.66 | 0.81 | 2.1 | 7.4 | 16.1 |
| | Early-Stop Extrapolation | 10.46 | 0.66 | 0.81 | 3.7 | 8.8 | 17.8 |
| | Early-Dynamics Baseline | 7.33 | 0.78 | 0.88 | 21.5 | 37.7 | 48.6 |
| | ProxyLM | 6.76 | 0.86 | 0.93 | 30.9 | 45.8 | 62.5 |
| | **TUNEAHEAD (Full)** | **5.02** | **0.86** | **0.93** | **36.0** | **55.8** | **73.3** |
| Qwen2-0.5B | Early-Stop Extrapolation | 10.41 | 0.69 | 0.83 | 6.3 | 11.7 | 18.7 |
| | Domain-Proxy Baseline | 9.37 | 0.73 | 0.85 | 10.0 | 16.4 | 25.3 |
| | Early-Dynamics Baseline | 7.47 | 0.81 | 0.90 | 19.3 | 28.8 | 39.7 |
| | ProxyLM | 4.00 | 0.90 | 0.95 | 35.6 | 53.7 | 72.1 |
| | **TUNEAHEAD (Full)** | **3.75** | **0.91** | **0.95** | **39.2** | **58.4** | **74.6** |

variance; *Pearson $r$* measures linear correlation between $P_{i,j}$ and $R_{i,j}$; $Acc@kpp$ is the fraction of predictions with $|P_{i,j} - R_{i,j}| \leq k$ percent. We additionally provide per-domain breakdowns in Appendix A.7, calibration curves in Appendix A.8, and 95% CIs (bootstrap) with paired permutation tests for significance in Appendix A.9.

**Exp-1: Predicting Fine-Tuning Performance and Generalization.** In the first set of experiments, we establish the end-to-end predictive strength of **TUNEAHEAD** when both dataset properties and LoRA hyperparameters vary. This set of experiments is designed to validate the accuracy and generalizability of our predictor (G2).

We construct a meta-dataset of over 1,300 complete fine-tuning runs spanning heterogeneous instruction-tuning sources and typical LoRA settings. Unless otherwise specified, all experiments reported in the main text use Qwen2.5-7B-Instruct as the base LLM, with ground truth defined as the *seed-averaged* MMLU test accuracy from full fine-tunes. Curation and training details are provided in Appendix A. To further test generalizability across architectures and scales,

we also construct additional meta-datasets on Llama-3-8B and Qwen2-0.5B, experiment results reported in Table 2.

**TUNEAHEAD** *sets a new accuracy bar for this prediction task.* As Table 1 shows, on the held-out test set, it **cuts RMSE by 30%** relative to the strongest non-**TUNEAHEAD** baseline (2.11→1.47 , ProxyLM) and by **80%** relative to Early-Stop Extrapolation (7.43→1.47). Tight-tolerance accuracy improves markedly: **Acc@1pp** +9.3 points (**+22.9%** rel.), **Acc@2pp** +14.6 points (**+21.5%**), and **Acc@3pp** +9.3 points (**+10.8%**) over the best baseline, while maintaining near-perfect ranking correlation (Pearson = 0.99). These gains reflect the *complementarity* of static data descriptors and dynamic 100-step probe signals: either source alone yields ∼ 3.4–3.5 RMSE, whereas their combination reaches **1.47** (**-56%** vs. dynamic-only; **-58%** vs. static-only). Figure 4 visually corroborates these numerical results. The plot for **TUNEAHEAD** (Full) exhibits the tightest clustering of points along the diagonal, confirming its superior accuracy. In contrast, even the strongest baseline, ProxyLM, shows larger deviations, particularly at the tails of the distribution.

**Generalization Across Architectures and Scales.** A critical question is whether **TUNEAHEAD**'s predictive power is confined to the Qwen2.5-7B-Instruct model. To proactively assess this, we constructed two additional, smaller-scale meta-datasets using Llama-3-8B (different architecture, 400 runs) and Qwen2-0.5B (smaller scale, 450 runs). As shown in Table 2, **TUNEAHEAD** continues to capture predictive signal in both cases, achieving $R^2 = 0.86$ on Llama-3-8B and $R^2 = 0.91$ on Qwen2-0.5B, with reasonably high Acc@kpp. Although RMSE is higher, accuracy is lower than in our primary experiments (due to the significantly smaller meta-datasets). These results indicate that the **TUNEAHEAD** feature design remains informative beyond the primary Qwen2.5-7B-Instruct setting. However, the predictors in Table 2 are trained within each target setting, and the smaller Llama-3-8B and Qwen2-0.5B meta-datasets naturally lead to higher errors than the primary 1,300-run setting. We therefore interpret these results as evidence of framework portability rather than evidence of universal zero-shot transfer.

**Generalization Across Benchmarks and Tasks.** To mitigate concerns that our features might overfit MMLU, we re-evaluate all 1,300+ fine-tuned checkpoints on **TruthfulQA**. Using the *exact same* meta-features and training protocol, **TUNEAHEAD** achieves consistently low prediction error and high $R^2$ on TruthfulQA as well (Table 3), outperforming zero/low-cost baselines (ProxyLM, Early-Stop Extrapolation, Early-Dynamics Baseline). Beyond benchmarks, we also build a **Summarization** mini meta-dataset (on Samsum (Gliwa et al., 2019), with ROUGE-L Score (Lin, 2004)) and observe similarly strong predictability; full results and implementation details are in **Appendix Table E.1**.

*Table 3.* **TruthfulQA (MC2) prediction quality.** Errors are in percentage points (pp). MC2 ACC@ $k$pp is the fraction of test cases with absolute prediction error $\leq k$ pp.

| Method | RMSE↓ | $R^2$ ↑ | $r$ ↑ | MC2 ACC@ $k$pp↑ | |
| --- | --- | --- | --- | --- | --- |
| | | | | $k=3$ | $k=5$ |
| Domain-Proxy Baseline | 6.33 | 0.85 | 0.92 | 37.0 | 53.8 |
| Early-Stop Extrapolation | 7.10 | 0.83 | 0.91 | 38.4 | 61.1 |
| TuneAhead-Static-Only | 3.90 | 0.95 | 0.98 | 48.6 | 64.6 |
| TuneAhead-Dynamic-Only | 3.60 | 0.95 | 0.98 | 52.3 | 71.7 |
| Early-Dynamics Baseline | 3.45 | 0.96 | 0.98 | 62.5 | 73.7 |
| ProxyLM | 2.54 | 0.97 | 0.99 | 70.6 | 84.4 |
| **TuneAhead (Full)** | **2.17** | **0.98** | **0.99** | **74.8** | **88.2** |

*Table 4.* Go/no-go deployment trade-off under different success thresholds. Stricter thresholds send fewer runs to full fine-tuning and save more compute, but also retain fewer successful runs.

| Threshold | True success rate | Runs sent to FT | Net saving | Successes retained |
| --- | --- | --- | --- | --- |
| 50% | 44.0% | 43.4% | 51.6% | 96.0% |
| 55% | 38.2% | 36.1% | 58.4% | 94.5% |
| 60% | 30.3% | 27.6% | 67.4% | 91.3% |
| 65% | 19.1% | 17.7% | 77.3% | 88.0% |
| 70% | 2.9% | 8.8% | 86.2% | 84.0% |

**From prediction to screening.** Although **TUNEAHEAD** is trained and evaluated as a continuous predictor, practitioners often use such predictions to decide which runs should receive the full fine-tuning budget. We instantiate this decision rule with a pre-specified success threshold $\tau = 55\%$ in our main analysis, without tuning it on the held-out test split. Since $\tau$ is applied only after prediction, changing it does not affect RMSE, $R^2$, Pearson's $r$, or Acc@$k$pp; it only changes the deployment trade-off between saved compute and retained successful runs. Table 4 shows this trade-off under several thresholds. "True success rate" is the fraction of held-out runs whose ground-truth score exceeds the threshold. "Runs sent to FT" is the fraction selected for full fine-tuning by the screening policy. "Net saving" accounts for the avoided full fine-tuning cost after probe overhead. "Successes retained" is the fraction of truly successful runs still selected for full fine-tuning.

These results show that compute saving is an operating-point-dependent deployment outcome rather than an intrinsic property of the predictor itself; we therefore treat continuous prediction metrics as the primary measure of **TUNEAHEAD**'s predictive quality.

*Summary.* (i) Heuristics based solely on early learning curves are *not* sufficient for high-precision prediction; (ii) dataset-intrinsic signals provide complementary, non-redundant information; (iii) their integration in **TUNEAHEAD** yields **low-error, tightly calibrated** predictions (95.1% within ±3pp), which is the regime practitioners care about for reliable pre-screening without full fine-tuning; (iv) preliminary cross-model experiments suggest that the **TUNEAHEAD** framework remains portable across different model scales and architectures.

**Exp-2: Diagnosis.** Beyond predictive accuracy, a core

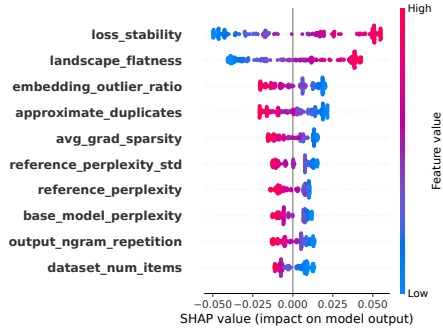

*(a)* Global feature importance (SHAP summary).

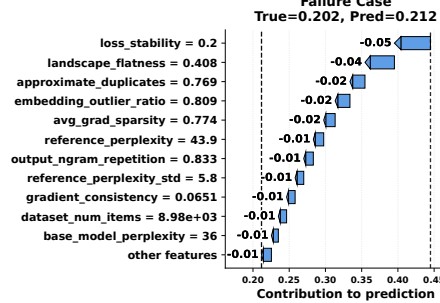

*(b)* Failure case ($E[f(X)] = 0.45$, $f(x) = 0.212$).

*Figure 5.* (a) SHAP summary plot ranking the global importance of meta-features for predicting fine-tuning success; (b) SHAP waterfall plot for a representative failure case (the model correctly predicted low performance).

contribution of **TUNEAHEAD** is its ability to provide diagnostic insights (G3). In this section, we use TreeSHAP to analyze the trained model and understand the key drivers of fine-tuning success or failure.

**Global Feature Importance**. The SHAP analysis in Figure 5(a) identifies the most influential features of fine-tuning performance. *Loss stability* and *landscape flatness* emerge as the most critical factors, confirming that a stable initial learning phase is paramount. Following these, data quality metrics like *embedding outlier ratio* and *approximate duplicates* are the strongest negative predictors, quantifying the high cost of noisy data. We trained predictors under multiple random seeds and computed TreeSHAP attributions for each. While the exact ranking of features varied slightly across seeds, the same core set of top features consistently emerged, with only minor shifts in their weights.

**Case Study: Diagnosing a Predicted Failure Run.** While the summary plot reveals global trends, **TUNEAHEAD**'s utility shines in diagnosing individual runs. To demonstrate its practical value, we conduct an in-depth analysis of a failure case, which our model correctly predicted.

The SHAP plot Figure 5(b) explains how meta-features contribute to pushing the performance prediction from the average prediction ($E[f(X)] = 0.45$) down to its final low score ($f(x) = 0.212$).

The diagnosis reveals a multi-faceted failure, jointly driven by poor learning dynamics and low data quality. On the dynamics side, *loss stability* is extremely low (0.20), whereas successful runs typically range from 0.5 to 0.8; values below 0.3 almost always fail, indicating an unstable trajectory. *Landscape flatness* is also poor (0.408), compared to successful runs clustering around 0.6, with values below 0.5 linked to sharp minima and poor generalization. On the data side, the duplicate rate is 0.769, far above the successful-run median of 0.387. The *embedding outlier ratio* is similarly extreme at 0.809, while successful runs typically stay below 0.5 (median 0.413).

This diagnosis provides a set of targeted, evidence-based prescriptions for the practitioner: **(1) Optimization instability**: Lower the learning rate or adjust optimizer hyperparameters to encourage stable convergence, and **(2) Data quality defects**: Apply semantic de-duplication to reduce redundancy and use embedding-based outlier removal to clean the dataset before fine-tuning.

As a proof of actionability, we applied simple adjustments guided by the diagnosis. Concretely, on the model side, we reduced the learning rate from $3 \times 10^{-5}$ to $1 \times 10^{-5}$ and adjusted the optimizer hyperparameters by lowering the AdamW momentum parameters ($\beta_1$ from 0.9 to 0.85 and $\beta_2$ from 0.999 to 0.98), which encourages more stable convergence. On the data side, we applied SemDeDup (Abbas et al., 2023) for semantic de-duplication, which removes near-duplicates within a cosine threshold of 0.95, and performed embedding-based outlier removal by filtering out samples whose sentence embeddings lie beyond 3 standard deviations from the mean in the representation space. These adjustments improved the run's final MMLU score significantly, from 20.2% to 48.7%.

**Exp-3: Ablation Study.** To validate our core design choices, we conduct two key ablation studies.

**(i) Static vs. Dynamic Feature Ablation.** This ablation validates our central hypothesis on the synergy between static and dynamic features. While **Exp-1** established the superior performance of the full model, a deeper look at the ablation results in Table 1 and Figure 4 reveals the nature of this synergy. These results show that the hybrid representation improves reliability by capturing complementary failure modes.

To evaluate this synergy, we partitioned runs into buckets (Table 5) based on whether the subset predictors succeeded (with Acc@2pp metric). The full model demonstrated robust improvements across all scenarios: (*i*) When only the dynamic predictor succeeded, the full model boosted Acc@2pp from 38.7% to 75.0%; (*ii*) conversely, when only the static predictor succeeded, it raised Acc@2pp from 37.3% to a perfect 100.0%. (*iii*) Even when both subset pre-

*Table 5.* Complementarity analysis on the test set. We report RMSE↓ and Acc@2pp↑; Buckets partition runs by whether static-only and dynamic-only predictions are correct.

| (i) Rescued-by-Dynamic | | | (ii) Rescued-by-Static | | |
|---|---|---|---|---|---|
| Model | RMSE | Acc@2pp | Model | RMSE | Acc@2pp |
| **TUNEAHEAD** (Full) | **1.43** | **75.0** | **TUNEAHEAD** (Full) | **0.69** | **100.0** |
| Static-only | 4.90 | 0.0 | Static-only | 2.92 | 37.3 |
| Dynamic-only | 3.22 | 38.7 | Dynamic-only | 4.60 | 0.0 |

| (iii) Both-subsets-correct | | | (iv) Both-subsets-wrong | | |
|---|---|---|---|---|---|
| Model | RMSE | Acc@2pp | Model | RMSE | Acc@2pp |
| **TUNEAHEAD** (Full) | **1.49** | **82.1** | **TUNEAHEAD** (Full) | **1.61** | **70.2** |
| Static-only | 3.34 | 36.2 | Static-only | 4.63 | 12.5 |
| Dynamic-only | 3.16 | 40.6 | Dynamic-only | 5.09 | 12.5 |

dictors succeeded, the full model still improved Acc@2pp from ∼40% to 82.1%, (*iv*) and when both failed, it dramatically mitigated errors, improving Acc@2pp from 12.5% to 70.2%. These rescue effects confirm that static and dynamic features capture different failure modes (data-level flaws vs. model-level instabilities), and their combination is the key to achieving robust prediction.

**(ii) Impact of Probe Budget.** A key hyperparameter in our framework is the length of the dynamic probe run. To justify our choice of 100 steps, we examine the trade-off between prediction accuracy and the computational cost of the probe. As illustrated in Figure 6, a clear "elbow point" emerges around the 100-step mark, indicating a point of diminishing returns. Specifically, increasing the probe budget from 0 to 100 steps yields a significant improvement in accuracy, with RMSE dropping from 3.50 to 1.47 and Acc@2pp rising from 32.8% to 82.5%. However, extending the budget further to 200 steps provides only a negligible gain, with Acc@2pp improving by just 0.3 percentage points to 82.8%. This minimal accuracy improvement comes at a substantial cost, as the average probe time increases linearly, nearly rising from 7.76 minutes to 12.11 minutes. Therefore, we select the 100-step probe as the default configuration in the present LoRA-based PEFT setting, where it offers a strong balance between predictive accuracy and the computational overhead mandated by our design goals (G1).

However, we do not view this value as universal: larger models, longer-context tasks, full-parameter fine-tuning, or substantially different optimization regimes may shift the optimal probe budget. This motivates adaptive probing strategies that stop once predictive uncertainty, decision margins, or dynamic statistics have stabilized.

# 6. Conclusion and Future Work

**Conclusion.** We introduced **TUNEAHEAD**, a pre-hoc framework for predicting fine-tuning performance before committing to full training. By combining static dataset descriptors with dynamic signals from a short probe run, **TUNEAHEAD** provides accurate continuous performance estimates and SHAP-based diagnostic attributions. Across

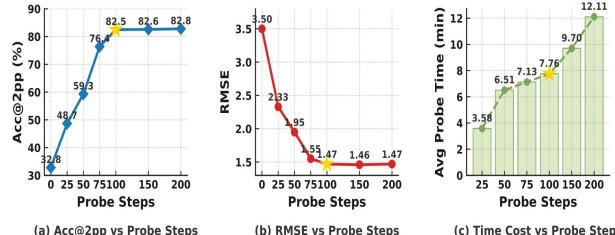

(a) Acc@2pp vs Probe Steps  (b) RMSE vs Probe Steps  (c) Time Cost vs Probe Steps

*Figure 6.* Effect of probe length on prediction accuracy, stability, and time cost. (a) Accuracy at 2pp steadily improves with longer probe runs but exhibits diminishing returns beyond 100 steps. (b) RMSE decreases sharply in the early stage and stabilizes after 100 steps. (c) Average probe time cost grows near-linearly with probe length, with 200 steps requiring about 1.5x the cost of 100 steps.

more than 1,300 fine-tuning runs, **TUNEAHEAD** consistently outperforms proxy-model, early-extrapolation, and single-view ablation baselines, showing that fine-tuning decisions can be made more resource-aware and data-driven.

**Limitations.** The current study has several limitations. First, a trained **TUNEAHEAD** predictor is not yet universal across model families, scales, and task distributions. Although the framework performs well on several base models, dynamic probe features are tied to the optimization behavior of the underlying model and may change across architectures or training regimes. In practice, this means that **TUNEAHEAD** should be viewed as a reusable framework rather than a single fixed predictor that transfers unchanged to every new model. For a substantially different base model, a small target-specific meta-dataset or calibration set may still be needed to recover reliable predictions. Second, our experiments focus on LoRA-based fine-tuning under a modest predefined hyperparameter space; generalization to QLoRA, other PEFT variants, full-parameter fine-tuning, or wider hyperparameter sweeps remains to be validated. Third, the 100-step probe is an empirical default in our setting rather than a universal budget. Finally, compute-saving numbers should be interpreted as deployment-policy outcomes, since they depend on the candidate-pool success rate, the chosen threshold, the predictor's screening behavior, and the probe overhead.

**Future Work.** Future work can improve **TUNEAHEAD** along three directions. The first is lightweight transfer and calibration: instead of collecting a full meta-dataset for every new base model, it is important to study which meta-features remain stable across related models and how many calibration runs are needed for reliable adaptation. The second is adaptive probing, where the system stops early when the go/no-go decision is already confident and continues probing when early signals remain unstable. The third is closing the loop between prediction, diagnosis, and intervention, extending the current dataset/run-level screening framework toward finer-grained data selection and automated dataset refinement.

## Acknowledgements

This work is supported by Guangdong provincial project (Project No. 2023CX10X008).

## Impact Statement

This work aims to improve the efficiency and reliability of fine-tuning large language models by predicting performance outcomes before committing to full fine-tuning. A primary positive impact is the reduction of wasted computational resources, which can lower financial and environmental costs and make large-scale model adaptation more accessible to researchers and practitioners with limited budgets. By providing diagnostic signals rather than a single opaque score, our framework may also encourage more principled, data-centric development practices.

At the same time, the proposed method could be misused to accelerate or scale fine-tuning for harmful or misleading applications if applied without appropriate safeguards. Because our predictor operates only on dataset statistics and short probe signals, it does not assess the downstream intent, factuality, safety, or societal impact of a fine-tuning task. As such, TUNEAHEAD should not be viewed as a safety filter or as an endorsement of any particular fine-tuning objective, but rather as a technical decision-support tool whose effects depend on how it is deployed.

We emphasize that performance prediction and compute optimization should be complemented by human judgment, application-level risk assessment, and appropriate safeguards. In future work, integrating data quality diagnostics with content filtering or safety-aware criteria could help ensure that efficiency gains do not inadvertently facilitate harmful model use.

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

# A. Additional Experimental Details

## A.1. Meta-Dataset Curation

**Base Models and Tasks.**  Unless otherwise specified, our main meta-dataset is constructed on **Qwen2.5-7B-Instruct** and evaluated on **MMLU**, chosen for its broad coverage of knowledge and reasoning across domains. To assess out-of-distribution robustness, we additionally report results on **TruthfulQA** (MC2 accuracy) and **SAMSum** (ROUGE-L) using the same meta-feature pipeline, and we include cross-model experiments on **Llama-3-8B-Instruct** and **Qwen2-0.5B** to study generalization across architectures and scales.

**Dataset Collection** $\{D_i\}$**.**  We use public instruction-tuning sources (*e.g.,* Alpaca, Dolly) and both programmatically generated variants and LLM-synthesized examples seeded by these sources, yielding over 1,300 dataset versions. Controlled transformations include: (1) **Sub-sampling**: sizes 500–25k; (2) **Domain slicing**: STEM vs. humanities, etc.; (3) **Noise injection**: label noise 0–20%. These procedures induce wide variation in statistical, semantic, and structural properties.

**Hyperparameter Space** $\{H_j\}$**.**  For each $D_i$, we sweep LoRA hyperparameters: learning rate $\in \{1\mathrm{e}{-}5, 2\mathrm{e}{-}5, 3\mathrm{e}{-}5\}$, batch size $\in \{8, 16, 32\}$. This space covers common practitioner settings and captures realistic data–hyperparameter interactions.

**Ground Truth Protocol.** For every $(D_i, H_j)$, we run full LoRA fine-tuning to convergence and evaluate on the corresponding downstream benchmark to obtain $R_{i,j}$. Depending on the experimental setting, this benchmark is MMLU (accuracy), TruthfulQA (MC2 accuracy), or SAMSum (ROUGE-L). Each $R_{i,j}$ is the mean over 3 random seeds to reduce stochasticity.

## A.2. Meta-Feature Extraction

We compute a feature vector $V_{i,j}$ per experiment.

**Static features.**  Dataset-intrinsic signals (e.g., semantic diversity, label balance, and length statistics). Semantic embeddings use `all-MiniLM-L6-v2` for efficiency/quality trade-off. Reference Perplexity is computed under the frozen base model used in each experiment (e.g., Qwen2.5-7B-Instruct, Llama-3-8B, or Qwen2-0.5B).

**Dynamic probe features.**  A standardized 100-step probe run per experiment with AdamW and a linear scheduler. We log losses and gradients at each step and compute early-optimization descriptors.

**Preprocessing.**  We standardize the full feature matrix by z-score normalization before model fitting.

**Sensitivity to Embedding Model Choice.** We deliberately selected all-MiniLM-L6-v2 (22M parameters, $d = 384$) to minimize overhead. However, practitioners might prefer larger models (e.g., e5-large-v2, $\sim 335$M parameters, $d = 1024$) for higher semantic resolution.

While scaling up the embedding model increases the specific extraction cost, our analysis shows the impact on the total framework efficiency is negligible. The cost consists of two parts:

*Inference latency.* This scales with parameter count. Using a large model (e.g., 335M parameters) would increase inference time by approximately $10\times$–$15\times$ compared to MiniLM.

*Distance calculation.* This scales linearly with dimension $d$ (complexity $O(N^2 \cdot d)$). Increasing $d$ from 384 to 1024 increases calculation time by approximately $2.7\times$.

*Impact assessment.* Despite these increases, static feature extraction remains a one-time, offline process. Even with a large embedding model, extracting features for a standard dataset (e.g., 10k samples) typically takes only a few minutes on a GPU. In contrast to the iterative fine-tuning probe (which runs forward and backward passes on the multi-billion-parameter target model), the embedding extraction cost constitutes $< 1\%$ of the total pipeline budget regardless of the model choice. Thus, the **TUNEAHEAD** framework is robust to the choice of embedding architecture.

## A.3. SHAP-Guided Feature Selection

**Setup.** We train a LightGBM predictor $F_\theta$ on a candidate feature set $\mathcal{F}$ and compute SHAP values on the validation set $D_{\text{val}} = \{x_i\}_{i=1}^n$. SHAP ensures an additive decomposition:

$$F_\theta(x_i) = \phi_{i,0} + \sum_{f \in \mathcal{F}} \phi_{i,f}.$$

For each feature $f$, we summarize its global contribution by the mean absolute SHAP and a direction-consistency statistic:

$$s_f := \frac{1}{n} \sum_{i=1}^n |\phi_{i,f}|, \qquad \rho_f := \text{SpearmanCorr}(x_{i,f}, \phi_{i,f}),$$

and encode the hypothesized sign by $\eta_f \in \{+1, -1\}$ (whether larger $x_{i,f}$ should increase or decrease the prediction). We use $c_f := \eta_f \rho_f$ for signed consistency.

**Step 1: Preliminary filtering.** We start from over 50 candidates (30 static features and 27 dynamic features). Through small ablations and sanity checks, we remove obviously weak or duplicate descriptors to obtain a screened pool $\mathcal{F}_1$.

**Step 2: SHAP value computation.** We fit $F_\theta$ on $\mathcal{F}_1$ and compute SHAP values $\{\phi_{i,f}\}$ on $D_{\text{val}}$ via `TreeExplainer`. We inspect beeswarm and bar plots to understand global effects.

**Step 3: Global contribution analysis.** We retain features that are both strong and consistent with theory. Concretely, we keep $f$ if

$$s_f \geq Q_{0.15}(\{s_f\}) \quad \text{and} \quad c_f := \eta_f \rho_f \geq 0.20,$$

and the Spearman correlation passes significance testing ($p < 0.05$). Here $Q_{0.15}$ denotes the 15th percentile of the empirical $\{s_f\}$.

**Step 4: Iterative pruning with CV safeguard.** From the retained set, we perform backward elimination. At each round, we remove the weakest candidate (smallest $s_f$ or negative $c_f$), retrain $F_\theta$, and accept the removal only if the cross-validated error does not worsen beyond a fixed tolerance:

$$\Delta\text{RMSE}_{\text{cv}} = \text{RMSE}_{\text{cv}}^{\text{new}} - \text{RMSE}_{\text{cv}}^{\text{old}} \leq \varepsilon, \qquad \varepsilon = 0.01.$$

Optionally, we use a robust tolerance tied to fold variability:

$$\varepsilon = \min(\text{SE}(\Delta), 0.01), \quad \text{SE}(\Delta) = \text{sd}(\{\Delta_k\}_{k=1}^K)/\sqrt{K},$$

and additionally require no significant degradation via a paired test (t-test or Wilcoxon), with $p \geq 0.05$. We stop when no feature can be dropped without violating the criterion.

**Outcome.** This SHAP-guided pipeline yields a compact, non-redundant meta-feature vector (14 static and 10 dynamic in **TUNEAHEAD**), balancing informativeness and stability while aligning with theoretical expectations.

## A.4. Prediction Model Training and Evaluation

**Training and Evaluation Details.** We split the 1,300+ fine-tuning experiments into 28% test, with the remaining 72% further divided into train/validation/calibration subsets (46/14/12%). Unless otherwise noted, the split uses a fixed random seed (=36). The predictor is a LightGBM gradient-boosted decision tree (GBDT) with the following hyperparameters fixed across all experiments: `learning_rate=0.05`, `num_leaves=4`, `n_estimators=140`, `subsample=0.6`, `colsample_bytree=0.6`, `min_child_samples=20`, and $\ell_2$ regularization `lambda_l2=1.0`. We use early stopping with a patience of 50 rounds. All models are trained with `n_jobs=-1` (multi-threading enabled).

For ablation studies, we define static-only features as dataset descriptors (e.g., length, lexical diversity, and perplexity), and dynamic-only features as probe-derived signals (e.g., loss decay and gradient variance). The full list of features is provided in Appendix B.

## A.5. Baselines

All baselines are evaluated under the same protocol across different base models and benchmarks when applicable.

**Ablation variants.** TUNEAHEAD-Static-Only (only static features) and TUNEAHEAD-Dynamic-Only (only dynamic probe features).

**Practical and literature-inspired baselines.** (1) *Early-Stop Extrapolation*: linear extrapolation of the 100-step validation loss (Domhan et al., 2015; Adriaensen et al., 2023); (2) *Loss-Rate Features*: rates of loss decrease under different schedules (Luo et al., 2025); (3) *Reference Perplexity*: reference-PPL as dataset difficulty proxy (Gururangan et al., 2020; Harada et al., 2025); (4) *ProxyLM* (Anugraha et al., 2024): regress on proxy-model scores (e.g., SmolLM-135M/360M, BLOOMZ-560M) optionally combined with dataset features.

## A.6. Evaluation Metrics and Protocols

We evaluate TUNEAHEAD with four standard regression and tolerance-based metrics:

- **RMSE (percentage points)**: measures absolute error between predicted and ground-truth performance,

$$\text{RMSE} = \sqrt{\frac{1}{N} \sum_{i,j} (P_{i,j} - R_{i,j})^2}.$$

- $R^2$: coefficient of determination, quantifying explained variance in ground-truth performance,

$$R^2 = 1 - \frac{\sum_{i,j}(R_{i,j} - P_{i,j})^2}{\sum_{i,j}(R_{i,j} - \bar{R}_{i,j})^2}.$$

- **Pearson** $r$: measures linear correlation between predictions and ground-truth,

$$r = \frac{\sum_{i,j}(P_{i,j} - \bar{P})(R_{i,j} - \bar{R})}{\sqrt{\sum_{i,j}(P_{i,j} - \bar{P})^2 \sum_{i,j}(R_{i,j} - \bar{R})^2}}.$$

- **Acc@$k$pp**: tolerance-based accuracy, defined as the fraction of predictions within $k$ percentage points of ground-truth,

$$\text{Acc@}k\text{pp} = \frac{1}{N} \sum_{i,j} \mathbf{1}\big(|P_{i,j} - R_{i,j}| \le k\big), \quad k \in \{1, 2, 3\}.$$

For all metrics, we average over three random seeds when obtaining ground-truth labels.

## A.7. Per-Domain Breakdown of Prediction Performance

To complement the aggregate results in Table 1, we report per-domain breakdowns of prediction accuracy on MMLU. The 57 subjects of MMLU are grouped into seven finer categories (STEM, Social Sciences, Humanities, Arts & Culture, Health & Medicine, Business & Professional, and Other/General Knowledge). This analysis verifies whether TUNEAHEAD consistently generalizes across heterogeneous domains.

As shown in Table A.1, the predictive performance of TUNEAHEAD is consistent across domains. RMSE varies only within $\pm 0.15$ across groups, and Pearson correlations remain above 0.98 throughout. Accuracy within $\pm 2$pp is also stable, ranging between 80–84%. This robustness indicates that the framework does not disproportionately benefit from or fail on particular subject categories, reinforcing its general applicability across diverse fine-tuning scenarios.

## A.8. Calibration Analysis

In Section 5, we emphasized that TUNEAHEAD's predictions are not only accurate in terms of correlation with fine-tuning outcomes, but also well calibrated in absolute values. This property is crucial for practical use cases, because a well calibrated predictor allows practitioners to directly interpret a predicted score as an approximate probability of fine-tuning success, enabling threshold-based go/no-go decisions without ad hoc post-processing.

*Table A.1.* Per-domain breakdown of prediction performance on MMLU. Domains are grouped into finer categories for clarity. Metrics include RMSE (pp), Pearson correlation $r$, and accuracy within $\pm2$pp tolerance.

| Domain | RMSE ↓ | $r$ ↑ | Acc@2pp ↑ |
|---|---|---|---|
| STEM (Math, CS, Physics, Bio) | 1.62 | 0.98 | 80.5 |
| Social Sciences (Econ, Psych, Soc) | 1.45 | 0.99 | 84.2 |
| Humanities (History, Philosophy, Law) | 1.53 | 0.99 | 83.1 |
| Arts & Culture (Literature, Linguistics, Art) | 1.59 | 0.98 | 81.9 |
| Health & Medicine (Clinical, Nutrition, Public Health) | 1.48 | 0.99 | 82.7 |
| Business & Professional (Mgmt, Exams) | 1.51 | 0.99 | 82.3 |
| Other / General Knowledge | 1.41 | 0.99 | 82.7 |
| Overall | 1.47 | 0.99 | 82.5 |

Figure A.1 shows the calibration curves of **TUNEAHEAD** compared with representative baselines. Each point corresponds to a bin of predicted scores, with the x-axis showing the mean predicted value and the y-axis showing the empirical success rate within that bin. The dashed line represents perfect calibration. We observe that several baselines (e.g., *Reference-PPL*, *Early-Stop*) systematically deviate from the diagonal, indicating a tendency to either overestimate or underestimate success probability. By contrast, **TUNEAHEAD**'s curve (red circles) remains consistently close to the diagonal across the full range, demonstrating superior calibration. This confirms that **TUNEAHEAD** is not only a strong ranker (as shown by the high correlations in Table 1), but also a reliable probability estimator, making its scores directly actionable for practitioners.

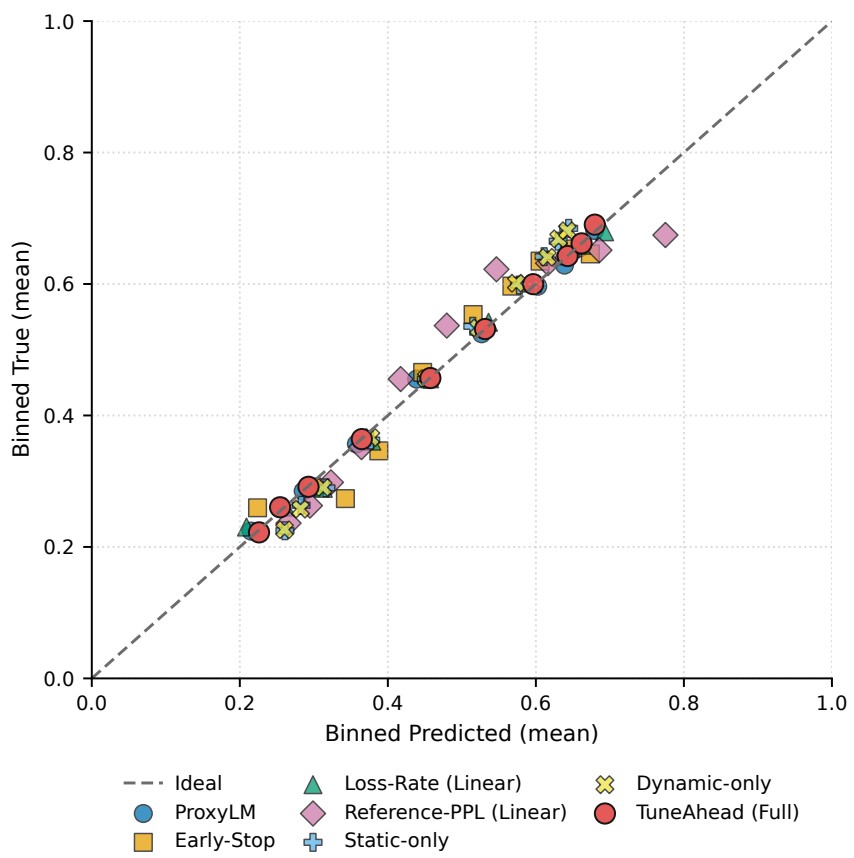

*Figure A.1.* Calibration plots of predicted scores across models. While baselines often deviate from the ideal diagonal, **TUNEAHEAD** (red) remains closely aligned with perfect calibration. This demonstrates that **TUNEAHEAD**'s predictions are both accurate in ranking and reliable in absolute probability estimation, complementing the aggregate results reported in Table 1.

*Table A.2.* Per-learning-rate performance of **TuneAhead** on the test set.

| Learning rate | RMSE ↓ | $R^2$ ↑ | $r$ ↑ | Accuracy (±kpp) ↑ | | |
|---|---|---|---|---|---|---|
| | | | | $k=1$ | $k=2$ | $k=3$ |
| $2 \times 10^{-4}$ | 1.36 | 0.99 | 1.00 | 51.68 | 85.23 | 96.64 |
| $3 \times 10^{-4}$ | 1.61 | 0.99 | 1.00 | 47.90 | 78.99 | 93.28 |

*Table A.3.* Per-batch-size performance of **TuneAhead** on the test set.

| Batch size | RMSE ↓ | $R^2$ ↑ | $r$ ↑ | Accuracy (±kpp) ↑ | | |
|---|---|---|---|---|---|---|
| | | | | $k=1$ | $k=2$ | $k=3$ |
| 8 | 1.58 | 0.99 | 1.00 | 51.96 | 76.47 | 92.16 |
| 16 | 1.47 | 0.99 | 1.00 | 45.56 | 84.44 | 96.67 |
| 32 | 1.32 | 0.99 | 1.00 | 52.63 | 88.16 | 97.37 |

## A.9. Confidence Intervals of Main Results

In Section 5, Table 1 reported the aggregate RMSE and accuracy of **TuneAhead** and representative baselines. While those results demonstrated clear performance gains, it is also important to examine the robustness of these findings under statistical resampling. To this end, we conducted bootstrap analysis ($N$=1000 resamples) and computed 95% confidence intervals for both RMSE and accuracy.

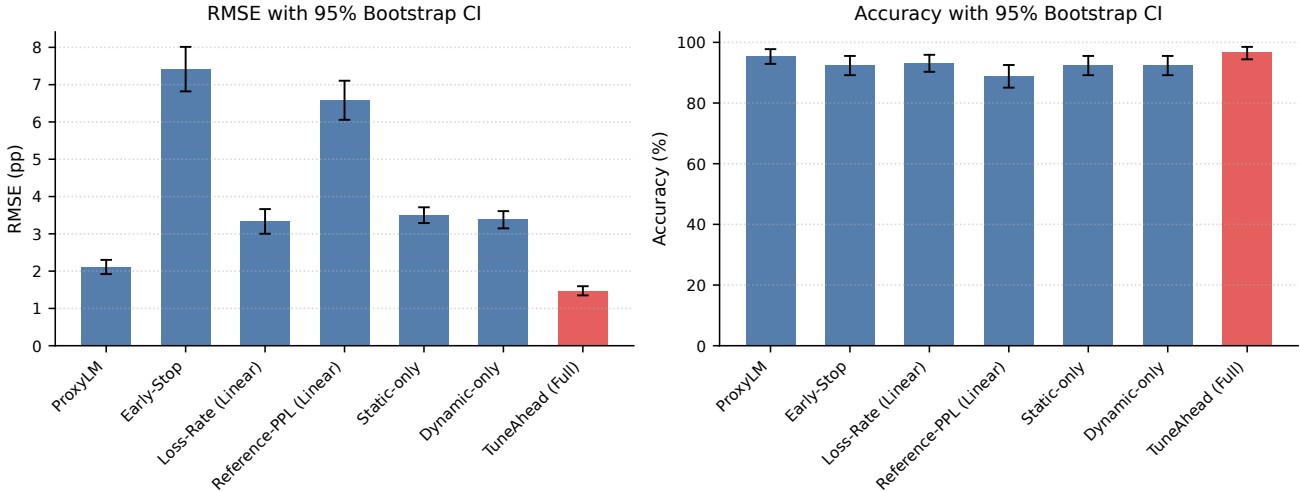

*Figure A.2.* 95% bootstrap confidence intervals for RMSE (left) and accuracy (right) across **TuneAhead** and baseline predictors. Red bars highlight **TuneAhead** (Full), while blue bars indicate baselines. The results confirm that **TuneAhead** not only achieves the lowest RMSE but also the highest accuracy, with narrow confidence intervals that do not overlap with most baselines.

As shown in Figure A.2, **TuneAhead** achieves the most stable performance: its RMSE is significantly lower than all baselines, and its accuracy is consistently higher. The narrow error bars demonstrate that these results are not due to random variation, but reflect a statistically robust advantage of combining static and dynamic signals.

## A.10. Per-Hyperparameter Breakdown

To verify that the improvements of **TuneAhead** are not tied to a narrow choice of optimization settings, we report test-set performance grouped by key hyperparameters as shown in the tables below.

Overall, the per-hyperparameter breakdown reinforces the robustness of **TuneAhead**. Across different learning rates (Table A.2), the predictor achieves consistently low RMSE (1.3–1.6pp) and nearly perfect correlation ($r \approx 1.0$). When grouping by batch size (Table A.3), larger batches generally yield slightly lower RMSE and higher calibration accuracy. Finally, the two-way grid (Table A.4) confirms that performance remains strong across all (learning rate × batch size)

*Table A.4.* Two-way breakdown by (learning rate × batch size).

| Learning rate | Batch size | RMSE ↓ | $R^2$ ↑ | $r$ ↑ | Accuracy (±kpp) ↑ | | |
|---|---|---|---|---|---|---|---|
| | | | | | $k{=}1$ | $k{=}2$ | $k{=}3$ |
| $2 \times 10^{-4}$ | 8 | 1.37 | 0.99 | 1.00 | 56.16 | 82.19 | 95.89 |
| $2 \times 10^{-4}$ | 16 | 1.32 | 0.99 | 1.00 | 50.00 | 87.50 | 97.50 |
| $2 \times 10^{-4}$ | 32 | 1.39 | 0.99 | 1.00 | 44.44 | 88.89 | 97.22 |
| $3 \times 10^{-4}$ | 8 | 2.03 | 0.98 | 0.99 | 41.38 | 62.07 | 82.76 |
| $3 \times 10^{-4}$ | 16 | 1.59 | 0.99 | 1.00 | 42.00 | 82.00 | 96.00 |
| $3 \times 10^{-4}$ | 32 | 1.27 | 0.99 | 1.00 | 57.50 | 85.00 | 95.00 |

combinations. These results demonstrate that **TUNEAHEAD**'s improvements are not contingent on narrow hyperparameter choices, but generalize broadly across optimization regimes.

## B. Meta-Feature Compendium

This appendix provides a detailed compendium of all candidate static and dynamic meta-features engineered for the **TUNEAHEAD** framework. For each feature, we include its **definition**, **formula** (when applicable), **acquisition method**, **predictive hypothesis (signal)**, and relevant literature references.

**Relation to data-centric learning.** The static feature families in TUNEAHEAD instantiate a broader data-centric view of model behavior: dataset size, redundancy, incompleteness, missingness, acquisition bias, and general data-quality pathologies can all affect the expected utility of training. Related work has studied these issues through data-quality characterization, coreset selection over incomplete data, selective data acquisition, cost-effective missing-value imputation, data-lake-based imputation under limited redundancy, and data generation under data scarcity (Fan et al., 2013; Chai et al., 2023; 2022; 2025; Yang et al., 2025b; Chai et al., 2024). In **TUNEAHEAD**, analogous signals are aggregated into run-level meta-features for predicting fine-tuning payoff.

## B.1 Notation for Meta-Feature Definitions

For clarity, we summarize the symbols used in the meta-feature definitions in Appendix B. Unless otherwise stated, all quantities are defined for a single dataset-hyperparameter pair $(D_i, H_j)$ and its associated probe run.

**Global objects.**

$D_i$ The $i$-th fine-tuning dataset (instruction–response pairs).

$H_j$ The $j$-th hyperparameter configuration (e.g., learning rate, batch size, LoRA rank).

$M_0$ The frozen base language model used for reference perplexity and the probe run.

$\theta_0$ Model parameters of $M_0$ before the probe fine-tuning.

$\theta_T$ Model parameters after $T$ probe steps (end of the probe run).

$\theta^*$ Final probe parameters; in practice we set $\theta^* = \theta_T$.

$T$ Number of probe optimization steps (typically $T = 100$).

$N$ Number of training examples in the current dataset $D_i$ (re-used in several formulas).

$\| \cdot \|_2$ Euclidean (L2) norm of a vector.

**Static feature notation (Appendix B.1).**

$\mu_{\mathbf{len}}$  Mean token length of examples in $D_i$.

$\sigma_{\mathbf{len}}$  Standard deviation of token lengths in $D_i$.

$x$  Input sequence (prompt) in an instruction–response pair $(x, y)$.

$y$  Output sequence (response) in an instruction–response pair $(x, y)$.

$E(s)$  Embedding vector of sample $s$ in the dataset, computed by a fixed encoder.

$\mu$  Mean embedding vector over all samples in $D_i$: $\mu = \frac{1}{N} \sum_s E(s)$.

$\sigma$  Standard deviation of embedding distances to the mean, i.e. the scalar standard deviation of $\{\|E(s) - \mu\|_2\}_s$.

**EOR**  Embedding Outlier Ratio, $\text{EOR} = \frac{1}{N} \big| \{s : \|E(s) - \mu\|_2 > 3\sigma\} \big|$.

$D_i$  (in $\text{PPL}(D_i, M_0)$) Same dataset as above, used to compute reference perplexity.

$x_n$  The $n$-th training example (usually a concatenated input–output sequence) in $D_i$.

$p_{M_0}(x_n)$  Probability of sequence $x_n$ under the base model $M_0$.

$\text{PPL}(D_i, M_0)$  Reference perplexity of dataset $D_i$ on $M_0$: $\text{PPL}(D_i, M_0) = \exp\big(-\frac{1}{N} \sum_{n=1}^{N} \log p_{M_0}(x_n)\big)$.

**Loss–dynamics notation (Appendix B.2.1).**

$L_t$  Training loss at probe step $t$ ($t = 1, \ldots, T$).

$L_0$  Initial loss at the very beginning of the probe ($t = 0$), used as Initial Loss.

$\bar{L}$  Mean loss over the probe trajectory: $\bar{L} = \frac{1}{T} \sum_{t=1}^{T} L_t$.

$\ell_t$  Log-loss at step $t$, defined as $\ell_t = \log L_t$.

$a, b$  Intercept and slope of the linear fit $\ell_t \approx a + bt$ obtained by Ordinary Least Squares; the slope $b$ is used as the Loss Decay (more negative $b$ means faster adaptation).

$\sigma_L^2$  Loss Stability, i.e. variance of the loss along the probe: $\sigma_L^2 = \frac{1}{T} \sum_{t=1}^{T} (L_t - \bar{L})^2$.

**Gradient-based notation (Appendix B.2.2).**

$g_t$  Gradient of the training loss with respect to the model parameters at probe step $t$.

$\|g_t\|_2$  L2 norm of the gradient at step $t$.

$\mu_g$  Mean gradient norm across the probe: $\mu_g = \frac{1}{T} \sum_{t=1}^{T} \|g_t\|_2$.

$\sigma_g^2$  Variance of gradient norms: $\sigma_g^2 = \frac{1}{T} \sum_{t=1}^{T} (\|g_t\|_2 - \mu_g)^2$.

$c_t$  Gradient Consistency at step $t$, defined as cosine similarity between consecutive gradients: $c_t = \dfrac{g_t \cdot g_{t+1}}{\|g_t\|_2 \|g_{t+1}\|_2}$.

$g_k$  $k$-th coordinate of a gradient vector $g$.

$\epsilon$  Small positive threshold used to decide whether a coordinate is "effectively zero".

$s$  Gradient Sparsity: fraction of near-zero gradient coordinates, $s = \dfrac{|\{k : |g_k| < \epsilon\}|}{|g|}$, where $|g|$ is the number of coordinates in $g$.

**Model-based notation (Appendix B.2.3).**

$\Delta\theta$ Parameter Change Norm during the probe: $\Delta\theta = \|\theta_T - \theta_0\|_2$.

$L(\theta)$ Training loss evaluated at parameters $\theta$.

$\delta$ Small random parameter perturbation used to probe the loss landscape around $\theta^*$.

$\Delta L$ Loss Landscape Flatness proxy: $\Delta L = L(\theta^* + \delta) - L(\theta^*)$.

$P_{\mathbf{baseline}}$ Performance of the baseline (pre-probe) model on an out-of-domain evaluation task, used in the catastrophic forgetting proxy.

$P_{\mathbf{probe}}$ Performance of the probed / partially fine-tuned model on the same out-of-domain task.

$\Delta P$ Catastrophic Forgetting Proxy: $\Delta P = P_{\text{baseline}} - P_{\text{probe}}$.

$h$ Activation vector in a hidden layer for a given input.

$|h|$ Number of units (coordinates) in the activation vector $h$.

$s_a$ Activation Sparsity: fraction of near-zero activations, $s_a = \dfrac{|\{u : |h_u| < \epsilon\}|}{|h|}$, where $h_u$ is the $u$-th coordinate of $h$ and $\epsilon$ is the same small threshold as above.

## B.2 Static Meta-Features

Static features are computed from the dataset $D_i$ prior to training, sometimes using the frozen base model $\mathcal{M}_0$.

### B.2.1 GLOBAL STATISTICS

- **Token Lengths (Mean & Std Dev).**

$$\mu_{\text{len}}(D_i) = \frac{1}{N}\sum_{j=1}^{N}|x_j|, \qquad \sigma_{\text{len}}(D_i) = \sqrt{\frac{1}{N}\sum_{j=1}^{N}\big(|x_j| - \mu_{\text{len}}(D_i)\big)^2}.$$

**Acquisition:** Compute from input/output token counts. **Signal:**Beyond surface complexity, token lengths also reflect task formulation style. Datasets with extremely short inputs often lack linguistic structure and yield unstable gradients, while very long instructions introduce compositional reasoning that can reveal early adaptation bottlenecks. We observe that datasets with high token-length variance tend to produce noisy probe loss curves, indicating that this feature reliably distinguishes inconsistent data sources or uneven annotation quality. (Vettoruzzo et al., 2023; Moghe et al., 2024).

- **Input–Output Length Ratio.**

$$r_{\text{io}}(D_i) = \frac{1}{N}\sum_{j=1}^{N}\frac{|y_j|}{|x_j|}.$$

**Acquisition:** Average of output-to-input length ratios. **Signal:** This ratio correlates with task paradigms: low ratios characterize compressive tasks such as summarization, while high ratios reflect elaborative tasks like question answering or code generation. Such structural differences shape model adaptation speed and generalization (Lin, 2004; Narayan et al., 2018).This ratio also serves as a proxy for semantic compression vs. semantic expansion. Large ratios typically indicate generative tasks where hallucination risks or stylistic variability appear, which increases gradient variance and lowers gradient consistency. The ratio is scale-free, inexpensive, and complements perplexity in signaling the "reasoning burden" required by the dataset.

- **Special Character & Code Ratio.**

  Let $\mathbb{1}_{sc}(t)$ be 1 if token $t$ is a special or code token (from a fixed list) and 0 otherwise. Let $\mathrm{Tok}(D_i)$ be all tokens in $D_i$. Then

  $$r_{sc}(D_i) = \frac{\sum_{t \in \mathrm{Tok}(D_i)} \mathbb{1}_{sc}(t)}{|\mathrm{Tok}(D_i)|}.$$

  **Acquisition:** Count of special/code tokens vs. total. **Signal:** Elevated ratios typically indicate domain-specific corpora (e.g., programming or formula-heavy data) or unclean web text. Such distributions require tailored tokenization or model adaptation, as shown in large-scale text-to-text transfer studies (Allamanis et al., 2018; Raffel et al., 2020). This feature further helps differentiate "format-structured" domains—HTML, JSON, LaTeX, or code—from natural language. These domains often interact poorly with models pretrained on mixed corpora, producing out-of-distribution token transitions visible in perplexity and loss-decay trends. High ratios also correlate with increased difficulty in the early probe because special tokens amplify embedding outlier probability.

- **Approximate Duplicates Ratio.**

  For each example $j$, let $z_j$ be its concatenated text (e.g., $z_j = x_j + y_j$). Let $\mathrm{Jacc}(z_j, z_k)$ be the Jaccard similarity between the token sets of $z_j$ and $z_k$, and fix a threshold $\tau \in (0, 1)$ (e.g., $\tau = 0.9$). The (conceptual) duplicate ratio is

  $$\mathrm{Dup}(D_i) = \frac{2}{N(N-1)} \sum_{1 \le j < k \le N} \mathbb{1}\big[\mathrm{Jacc}(z_j, z_k) \ge \tau\big],$$

  which we approximate in practice using MinHash/LSH instead of exhaustive pairwise comparisons.

  **Acquisition:** Identify near-duplicates via embedding similarity threshold $\tau$. **Signal:** High duplication reduces effective dataset diversity, amplifies memorization, and weakens generalization. Deduplication in LLM pretraining has been shown to improve downstream performance and reduce overfitting (Carlini et al., 2023; Lee et al., 2022). Duplicate-heavy datasets artificially inflate dataset size while reducing effective information density. Higher duplicate ratios are strongly associated with shallow loss decay and low parameter movement, because the probe repeatedly sees nearly identical gradients. MinHash allows efficient detection of semantic near-copies and avoids quadratic pairwise operations.

- **Embedding Outlier Ratio.**

  Let $s_j$ denote the text for example $j$ (e.g., input or input+output), and $e_j = E(s_j) \in \mathbb{R}^d$ its embedding. Define the embedding mean and (scalar) standard deviation as

  $$\mu_E = \frac{1}{N} \sum_{j=1}^{N} e_j, \qquad \sigma_E = \sqrt{\frac{1}{N} \sum_{j=1}^{N} \|e_j - \mu_E\|_2^2}.$$

  The outlier ratio is

  $$\mathrm{EOR}(D_i) = \frac{1}{N} \sum_{j=1}^{N} \mathbb{1}\big[\|e_j - \mu_E\|_2 > 3\sigma_E\big].$$

  **Acquisition:** Detect large deviations in embedding space. **Signal:** Outlier samples often correspond to mislabeled, noisy, or domain-shifted data. Their presence destabilizes optimization and can severely degrade model robustness. Removing outliers is a key step in modern dataset curation pipelines (Hendrycks et al., 2019; Northcutt et al., 2022; Dodge et al., 2021). EOR highlights whether the dataset contains idiosyncratic or adversarial examples that do not align with the dominant semantic manifold. Such outliers typically cause spikes in loss stability and suppress gradient consistency during the probe. The three-sigma threshold is a robust and interpretable choice: it captures semantic anomalies without being overly sensitive to moderate tail imbalance.

- **Dataset Size (Num Items)**

  $$\mathrm{Size}(D_i) = N,$$

  i.e., the number of examples in $D_i$.

  **Acquisition:** Dataset example count. **Signal:** Larger datasets typically improve model performance, but the gains follow a power-law with diminishing returns. Scaling law analyses show that optimal performance requires balancing dataset size with model capacity and compute budget(Kaplan et al., 2020; Hoffmann et al., 2022).

B.2.2 LEXICAL DIVERSITY FEATURES

- **Type–Token Ratio.**

  Let $T(D_i) = |\text{Tok}(D_i)|$ be the total number of tokens and $|V(D_i)|$ the vocabulary size. Then

  $$\text{TTR}(D_i) = \frac{|V(D_i)|}{T(D_i)}.$$

  **Acquisition:** Compute vocabulary diversity in $D_i$. **Signal:** A low TTR indicates repetitive content and limited lexical variety, which can impair a model's ability to generalize to unseen expressions. High TTR reflects lexical richness but may also introduce noise or rare tokens. TTR is a long-established measure of lexical richness and a standard meta-feature in dataset characterization (Rivolli et al., 2019).

- **N-gram Repetition.**

  Fix an $n$ (e.g., $n = 4$). Let $\mathcal{G}_n(D_i)$ be the multiset of all $n$-grams in $D_i$, and let $c(g)$ be the count of $g$ in $\mathcal{G}_n(D_i)$. We define the repetition ratio as

  $$\text{Rep}_n(D_i) = \frac{\sum_g \max\{c(g) - 1, 0\}}{\sum_g c(g)}.$$

  **Acquisition:** Fraction of repeated $n$-grams in outputs. **Signal:** High repetition rates often signal low-quality or synthetic outputs, reducing effective informational content and promoting degenerate training behavior. Low repetition may indicate dispersed data but could also reduce coherence. Repetition metrics are widely used in text degeneration and quality control studies(Rivolli et al., 2019; Holtzman et al., 2020).

- **Instruction Complexity.**

  Let $\text{depth}(x_j)$ denote the syntactic parse-tree depth (or another fixed complexity score) of input $x_j$. Then

  $$\text{IC}(D_i) = \frac{1}{N} \sum_{j=1}^{N} \text{depth}(x_j).$$

  **Acquisition:** Average parse-tree depth of instructions. **Signal:** Shallow trees suggest trivial instructions that under-challenge the model, while overly deep trees reflect high syntactic complexity that may hinder comprehension or stable learning. Balanced complexity encourages both learnability and generalization (Yatskar, 2019)

  Together, lexical richness and repetition patterns help differentiate high-entropy datasets from templated or pattern-locked corpora. Higher lexical diversity often correlates with smoother probe adaptation, whereas heavy $n$-gram repetition is a known predictor of overfitting in short-horizon training. Instruction complexity, approximated via parse-tree depth, is also predictive of whether the model must engage multi-step reasoning, which tends to manifest as slower yet more stable loss decay.

B.2.3 INFORMATION-THEORETIC PROPERTIES

- **Reference Perplexity.**

  Let $M_0$ be a frozen base model and $x_n$ the $n$-th token in a concatenated sequence from $D_i$ of length $N_{\text{tok}}$. The reference cross-entropy is

  $$\text{CE}(D_i, M_0) = -\frac{1}{N_{\text{tok}}} \sum_{n=1}^{N_{\text{tok}}} \log p_{M_0}(x_n \mid x_{<n}),$$

  and the reference perplexity is

  $$\text{PPL}(D_i, M_0) = \exp\big(\text{CE}(D_i, M_0)\big).$$

  We use both the mean and standard deviation of token-level losses as features.

  **Acquisition:** Measure using frozen base model. **Signal:** High perplexity indicates distributional mismatch between dataset and pretraining corpus, leading to slower convergence and increased adaptation cost. Low perplexity suggests greater alignment with prior knowledge. PPL remains a widely accepted proxy for domain mismatch and learning

difficulty(Wu et al., 2017; Jozefowicz et al., 2016). Beyond measuring difficulty, reference perplexity acts as a proxy for model surprise under zero adaptation. Datasets with high mean perplexity typically exhibit steeper early improvements (more negative loss-decay slopes), but at the cost of higher volatility due to stronger domain mismatch. Considering both the mean and the standard deviation of perplexity therefore captures not only overall shift but also internal heterogeneity within the dataset.

- **Input–Output Semantic Similarity (IO Similarity)**

  Let $e_j^x = E(x_j)$ and $e_j^y = E(y_j)$ be embeddings of input and output. We define

  $$\text{IO-Sim}(D_i) = \frac{1}{N} \sum_{j=1}^{N} \cos\big(e_j^x, e_j^y\big).$$

  **Acquisition:** Cosine similarity of embeddings. **Signal:** Low similarity may reflect irrelevant or hallucinated outputs, while very high similarity often indicates trivial paraphrasing lacking informativeness. Moderate levels of similarity are most effective for meaningful adaptation. Embedding-based similarity has been widely studied in representation learning (Clark et al., 2020; Cer et al., 2018).

- **Output Semantic Diversity.**

  Let $e_j^y = E(y_j)$ and define similarity $\text{Sim}(y_j, y_k) = \cos(e_j^y, e_k^y)$. Then

  $$\text{Div}(D_i) = \frac{2}{N(N-1)} \sum_{1 \le j < k \le N} \big(1 - \text{Sim}(y_j, y_k)\big).$$

  **Acquisition:** Average pairwise output dissimilarity. **Signal:** Low diversity signals redundancy and narrow coverage, while excessive diversity may indicate incoherence or noisy task signals. Balanced semantic diversity provides both robustness and coverage, promoting better generalization (Ziegler et al., 2020; Li et al., 2016).

  These semantic features measure representational compression of the task. High input–output similarity implies strongly input-grounded tasks, while low similarity or high output semantic diversity suggests creative or open-ended generation. In practice, they explain failure modes where probe loss decay appears strong, but final task performance is poor due to hallucination or uncontrolled style drift.

- **LM-Data Vocabulary Alignment (KL Divergence).**

  Let $V$ be a shared vocabulary. Let $P(w)$ be the smoothed unigram frequency of token $w$ in $D_i$ and $Q(w)$ the corresponding frequency in a reference pretraining corpus. Then

  $$\text{KL}(P \,\|\, Q) = \sum_{w \in V} P(w) \log \frac{P(w)}{Q(w)}.$$

  **Acquisition:** Compare dataset and reference corpus word frequencies. **Signal:** Large divergence highlights domain shift, suggesting that the dataset vocabulary departs from the pretraining distribution. This increases the adaptation burden and may reduce efficiency. KL divergence is a standard measure in domain adaptation and data selection (Aharoni & Goldberg, 2020; Axelrod et al., 2011).

- **Answer Groundedness.**

  For each example $(x_j, y_j)$, let $\text{ngrams}(y_j)$ and $\text{ngrams}(x_j)$ be the multisets of $n$-grams (for a fixed $n$). Define per-example groundedness

  $$g(y_j, x_j) = \frac{|\text{ngrams}(y_j) \cap \text{ngrams}(x_j)|}{|\text{ngrams}(y_j)|},$$

  and dataset-level groundedness

  $$G(D_i) = \frac{1}{N} \sum_{j=1}^{N} g(y_j, x_j).$$

**Acquisition:** Ratio of overlapping n-grams between output and input. **Signal:** Low groundedness suggests hallucination or irrelevant generation, while overly high groundedness may reduce informativeness by copying excessively. Moderate grounding balances fidelity with informativeness, ensuring both reliability and novelty (Ji et al., 2023; Zhao et al., 2020).

Vocabulary KL focuses on lexical mismatch that pure perplexity may blur, for example rare-domain terminology or shifting tokenization patterns. Answer groundedness, in contrast, quantifies how much the dataset encourages copying versus abstraction: very low groundedness correlates with hallucination-prone gradients, whereas excessively high groundedness correlates with shallow parameter updates that under-explore model capacity. These two features thus capture complementary aspects of domain shift and supervision style.

## B.3 Dynamic Probe Meta-Features

Dynamic features are extracted during a standardized 100-step probe run.

### B.3.1 LOSS-BASED INDICATORS

- **Initial Loss.**

$$L_0(D_i, H_j) = L(\theta_0; D_i),$$

  i.e., the probe loss evaluated at parameters $\theta_0$ before any updates (on a fixed probe batch or mini-epoch).

  **Acquisition:** Probe loss at the first optimization step. **Signal:** The initial loss measures how well the pretrained model aligns with the dataset before adaptation. High values suggest a significant domain gap, requiring more updates to adapt, while low values indicate better alignment and easier fine-tuning. It is widely used as a proxy for domain difficulty in transfer learning (Arpit et al., 2017; Hestness et al., 2017). Initial loss $L_0$ is particularly useful for detecting coarse domain mismatch. Datasets with extremely high $L_0$ often contain formatting inconsistencies or noisy annotations, which subsequently manifest as low gradient consistency and elevated catastrophic forgetting during the probe. Conversely, very low $L_0$ suggests that the base model is already well aligned with the task distribution.

- **Loss Decay Rate.**

  For step indices $t = 1, \ldots, T$, define log-loss $\ell_t = \log L_t$. We fit an Ordinary Least Squares (OLS) linear regression

$$\ell_t \approx a + bt$$

  by solving

$$(a^\star, b^\star) = \arg\min_{a,b} \sum_{t=1}^{T} (\ell_t - a - bt)^2.$$

  The Loss Decay feature is the slope

$$\alpha(D_i, H_j) = b^\star.$$

  More negative $\alpha$ indicates faster exponential decay of loss in the early training regime.

  **Acquisition:** Slope of regression fit on probe loss curve. LinReg means linear rgression with Ordinary Least Squares (OLS). Slope is **Signal:** A steep negative slope indicates strong learnability and rapid adaptation, whereas flat or unstable curves suggest noisy or hard-to-learn data. Early loss decay is highly predictive of final model performance (Wu et al., 2017; Hestness et al., 2017; Loog & Viering, 2022). The slope on the log-loss curve is deliberately chosen for its scale invariance and robustness to multiplicative noise. Small but consistently negative slopes indicate steady learning, while oscillatory or near-zero slopes typically correspond to datasets with noisy semantics or highly heterogeneous instruction formats. This formulation is also consistent with NTK-inspired (Jacot et al., 2020)linearization analyses, which predict approximately linear learning dynamics early in training.

- **Loss Curve Stability**

  Let

$$\bar{L} = \frac{1}{T} \sum_{t=1}^{T} L_t$$

be the mean probe loss. The (population) variance is

$$\sigma_L^2(D_i, H_j) = \frac{1}{T} \sum_{t=1}^{T} (L_t - \bar{L})^2.$$

**Acquisition:** Variance of loss during probe. **Signal:** Low variance implies stable optimization and smoother convergence, while high fluctuations often reflect noisy data, poor learning rates, or unstable alignment between model and task. Stable trajectories are associated with better generalization(Li et al., 2024; Wu et al., 2017). Loss volatility is a highly diagnostic signal for annotation noise, domain inconsistency, and unstable token distributions. Datasets with large $\sigma_L^2$ frequently show poor end-of-probe generalization and weaker gradient norms, indicating that the model is reacting to contradictory supervision. In contrast, low-volatility loss curves tend to correspond to well-curated dataset where the probe dynamics are smooth and predictable.

### B.3.2 GRADIENT-BASED INDICATORS

- **Gradient Norm (Mean & Variance)**

  Let $g_t = \nabla_\theta L_t$ be the gradient at step $t$ and $r_t = \|g_t\|_2$ its Euclidean norm. Define

  $$\mu_g(D_i, H_j) = \frac{1}{T} \sum_{t=1}^{T} r_t, \qquad \sigma_g^2(D_i, H_j) = \frac{1}{T} \sum_{t=1}^{T} (r_t - \mu_g(D_i, H_j))^2.$$

  **Acquisition:** Gradient norms across probe steps. **Signal:** Gradient norms reflect the strength of learning signals. Very large norms can cause instability or gradient explosion, while very small norms may stall learning or trap the model in poor local minima. Both mean and variance provide insight into learning dynamics (Pascanu et al., 2013; Sutskever et al., 2013; Killamsetty et al., 2021). The joint behavior of the mean and variance of $\|g_t\|_2$ encodes whether the dataset induces smooth or sharply fluctuating learning trajectories. Large variance is a hallmark of training instability and correlates with sharp minima detected by the flatness proxy, whereas consistently tiny norms suggest that the data provides little informative signal beyond the pretrained state. These statistics are cheap to obtain via hooks and yet strongly predictive of downstream calibration quality.

- **Gradient Consistency.**

  For $t = 1, \ldots, T-1$, define cosine similarity between consecutive gradients

  $$c_t = \cos(g_t, g_{t+1}) = \frac{\langle g_t, g_{t+1} \rangle}{\|g_t\|_2 \|g_{t+1}\|_2}.$$

  The consistency score is

  $$\mathrm{GC}(D_i, H_j) = \frac{1}{T-1} \sum_{t=1}^{T-1} c_t.$$

  **Acquisition:** Cosine similarity between sequential gradients. **Signal:** High consistency indicates coherent optimization paths, suggesting the model is learning a stable objective. Low or negative alignment suggests noisy or conflicting signals, slowing convergence. Gradient alignment is also linked to meta-learning generalization (Finn et al., 2017; Killamsetty et al., 2021; Guiroy et al., 2019). Gradient alignment offers a functional measure of how "well-posed" the supervision is. Datasets with conflicting signals—for example, heterogeneous styles or contradictory labels—lead to low cosine similarity between consecutive gradients and undermine the effectiveness of even strong learning rates. High gradient consistency, in contrast, indicates that updates point in broadly similar directions, and empirically correlates with both faster convergence and higher final accuracy.

- **Gradient Sparsity.**

  Fix a small threshold $\epsilon > 0$ (*e.g.*, $\epsilon = 10^{-6}$). Let $g_t \in \mathbb{R}^P$ and denote its coordinates by $g_t^{(k)}$. Define the fraction of near-zero coordinates at step $t$ as

  $$s_t = \frac{1}{P} \big| \{ k \in \{1, \ldots, P\} : |g_t^{(k)}| < \epsilon \} \big|.$$

The gradient sparsity feature is

$$\mathrm{GS}(D_i, H_j) = \frac{1}{T} \sum_{t=1}^{T} s_t.$$

**Acquisition:** Proportion of near-zero gradient components. **Signal:** High sparsity indicates that only a small subset of parameters is being updated, potentially limiting adaptation. Moderate sparsity may improve efficiency and generalization, but excessive sparsity may signal model–data mismatch (Frankle & Carbin, 2019; Evci et al., 2021; Killamsetty et al., 2021).Gradient sparsity measures how much of the parameter space is actively updated by the dataset. Highly sparse gradients indicate that only a small submanifold of parameters is being engaged, which is typical for overly templated or semantically shallow datasets. Probe runs with excessively sparse gradients almost always show small parameter movement and weaker downstream gains, highlighting under-utilization of model capacity.

**Relation to early-training data diagnostics.** The dynamic meta-features are also connected to recent data-centric methods that treat early training behavior as evidence about data utility or data defects. LEAD uses loss-, gradient-, and history-based signals for efficient instruction-data selection, while MisDetect uses early-loss behavior to identify potentially mislabeled examples (Lin et al., 2025b; Deng et al., 2024). TUNEAHEAD differs in objective and granularity: rather than selecting or cleaning individual instances, it aggregates early interaction signals to predict the run-level payoff of a dataset–hyperparameter pair.

### B.3.3 MODEL-BASED INDICATORS

- **Parameter Change Norm.**
$$\Delta\theta(D_i, H_j) = \|\theta_T - \theta_0\|_2,$$

where $\theta_0$ and $\theta_T$ are the parameter vectors before and after the $T$-step probe.

**Acquisition:** L2 norm of parameter changes during probing. **Signal:** Parameter change magnitude reflects adaptation strength. Moderate changes indicate healthy learning, while excessive shifts may reflect instability or overfitting. This feature has been used to analyze learning dynamics and compression strategies (Li et al., 2020; Raghu et al., 2017).This feature captures the effective update magnitude independently of the absolute loss scale. Under small learning-rate regimes, $\Delta\theta$ closely correlates with the local curvature of the loss landscape: extremely small values indicate underfitting or poor gradient informativeness, while excessively large values correlate with overshooting or movement into brittle minima. Moderate parameter change typically aligns with healthy, task-aligned adaptation.

- **Loss Landscape Flatness Proxy.**

Let $\theta^* = \theta_T$ denote the probe endpoint. Draw $K$ i.i.d. perturbations $\delta^{(k)} \sim \mathcal{N}(0, \sigma^2 I)$ with a fixed $\sigma > 0$, and define

$$\Delta L^{(k)} = L(\theta^* + \delta^{(k)}; D_i) - L(\theta^*; D_i), \qquad k = 1, \ldots, K.$$

The flatness proxy is the average loss increase

$$\Delta L(D_i, H_j) = \frac{1}{K} \sum_{k=1}^{K} \Delta L^{(k)}.$$

Smaller $\Delta L$ indicates a flatter local basin.

**Acquisition:** Apply perturbation $\delta$ and measure loss change. **Signal:** Flat minima (small $\Delta L$) correspond to more robust solutions with better generalization under distribution shifts, while sharp minima indicate overfitting and fragility. Flatness has been consistently linked to generalization performance(Wu et al., 2017; Li et al., 2024).We adopt perturbation-based flatness because Hessian-trace estimators are unstable and prohibitively expensive for large language models. The resulting $\Delta L$ reliably separates datasets that generalize well from those that heavily overfit to the short-horizon probe data. Empirically, lower $\Delta L$ values coincide with smoother loss curves and higher tolerance-based accuracy (e.g., Acc@2pp), supporting the classic link between flat minima and robust generalization.

- **Catastrophic Forgetting Proxy.**

Let $T_{\text{ood}}$ be a fixed out-of-domain evaluation task and $P(\theta; T_{\text{ood}})$ be its performance metric (e.g., accuracy). Define

$$P_{\text{baseline}} = P(\theta_0; T_{\text{ood}}), \qquad P_{\text{probe}} = P(\theta_T; T_{\text{ood}}).$$

The forgetting proxy is

$$\Delta P(D_i, H_j) = P_{\text{baseline}} - P_{\text{probe}}.$$

Larger $\Delta P$ indicates stronger interference with previous knowledge.

**Acquisition:** Performance drop on out-of-domain task during probe. **Signal:** Large drops indicate interference between new and old tasks, a hallmark of catastrophic forgetting. This proxy highlights whether fine-tuning data compromises existing knowledge(Wen & Itti, 2018). Although the probe uses only a small number of update steps, early forgetting is surprisingly predictive of final over-specialization or domain drift. Large $\Delta P$ values indicate that gradients induced by the new dataset interfere strongly with pretrained capabilities on unrelated tasks. This proxy therefore highlights data regimes where fine-tuning is likely to compromise core model knowledge, even if training loss appears to improve.

- **Activation Sparsity.**

  Consider a fixed hidden layer with activations $h_t \in \mathbb{R}^Q$ for a probe minibatch at step $t$ (flattening over tokens and units). Fix a threshold $\epsilon > 0$ and define

  $$s_t^{\text{act}} = \frac{1}{Q}\big|\{q \in \{1, \ldots, Q\} : |h_t^{(q)}| < \epsilon\}\big|.$$

  The activation sparsity feature is

  $$s_a(D_i, H_j) = \frac{1}{T}\sum_{t=1}^{T} s_t^{\text{act}}.$$

  **Acquisition:** Fraction of near-zero activations in hidden layers. **Signal:** Sparse activations suggest selective use of model capacity. Moderate sparsity improves interpretability and efficiency, while excessive sparsity indicates underutilized capacity or inefficient learning. It is widely used as a proxy for model resource utilization (Glorot et al., 2011; Maass, 1997).Activation sparsity approximates how widely the dataset engages the internal representation space of the model. Datasets with richer and more varied semantics tend to activate broader regions of the network, whereas narrow or highly templated datasets trigger sparse activations concentrated in predictable subspaces. This feature complements gradient sparsity by revealing a second level of representational bottleneck, and helps explain why some datasets yield limited transfer despite low training loss.

## C. Meta-Feature Computation Cost

Table C.1 reports the per-feature overhead for a typical dataset of 2.5k–5k examples. In this regime, static extraction completes in $\approx$1.6–3.2 minutes and dynamic, model-based proxies add $\approx$2.4–4.8 minutes, yielding a total of $\approx$4–8 minutes per dataset (excluding the probe run itself). Most static signals are computed in a single streaming pass over tokenized text and scale essentially linearly with corpus size; approximate-duplicate detection uses MinHash/LSH sketches to avoid quadratic pairwise comparisons; and semantic-diversity statistics rely on subsampling so that the effective complexity is closer to $O(N \log N)$ than $O(N^2)$ in practice. Model-based statics such as reference perplexity amortize well on a single GPU with fp16 inference, while the dynamic features are harvested "for free" from the 100-step probe via loss logs and forward/gradient hooks, so their marginal cost is negligible compared to the probe itself. Two design choices keep the end-to-end budget below 5% of a full fine-tune: static features are a one-time, offline cost that is cached at the dataset level, and dynamic features piggy-back on traces already produced during the probe.

**Relation to cost-aware LLM data workflows.** The cost analysis in Appendix C is also related to a broader line of cost-aware LLM data systems, where the goal is to reduce expensive model calls, prompting overhead, or data-preparation cost without sacrificing downstream quality. Examples include cost-effective in-context learning for entity resolution, weak-to-strong prompting for low-cost data transformation, natural-language data preparation, and LLM-based data-management pipelines (Fan et al., 2024; Li et al., 2025; Fan et al., 2025; Chen et al., 2023). TUNEAHEAD is complementary: instead of optimizing a deployed data workflow directly, it predicts whether a candidate fine-tuning run is worth executing.

In engineering the pipeline we found that a small number of implementation details materially reduce wall-clock time without degrading predictive quality. Tokenization, length histograms, n-gram counts, and vocabulary frequencies are computed once and reused across features such as TTR, repetition, KL divergence, and groundedness, eliminating redundant passes. For pairwise-style measures (near-duplicate ratio, output semantic diversity), we use sampling and sketching rather

| ID | Feature (Appendix) | Category | Computation (brief) | GPU | Typical cost | Notes |
|---|---|---|---|---|---|---|
| S1 | Token Lengths (Mean & Std) | B.1.1 | token length stats | No | 0.1–0.2 min | reuse tokenization |
| S2 | Input–Output Length Ratio | B.1.1 | avg $|y|/|x|$ | No | $< 0.1$ min | aggregation |
| S3 | Special Char & Code Ratio | B.1.1 | special/code tokens ratio | No | 0.1–0.2 min | regex/list |
| S4 | Approx. Duplicates Ratio | B.1.1 | MinHash/LSH near-dup | No | 0.3–0.6 min | subsampling |
| S5 | Embedding Outlier Ratio | B.1.1 | $3\sigma$ outliers in embed space | Opt. | 0.2–0.4 min | small-batch embed |
| S6 | Dataset Size (Num Items) | B.1.1 | count $N$ | No | $< 0.1$ min | — |
| S7 | Type–Token Ratio (TTR) | B.1.2 | unique/total tokens | No | 0.1–0.2 min | streaming |
| S8 | N-Gram Repetition Rate | B.1.2 | repeated n-grams in $y$ | No | 0.2–0.3 min | $n = 2/3$ |
| S9 | Instruction Complexity | B.1.2 | parse-tree depth avg | No | 0.2–0.4 min | light parser |
| S10 | Reference Perplexity (PPL) | B.1.3 | frozen $M_0$ forward PPL | Yes | 0.7–1.5 min | fp16/parallel |
| S11 | IO Semantic Similarity | B.1.3 | $\cos(E(x), E(y))$ | Opt. | 0.2–0.4 min | sampled pairs |
| S12 | Output Semantic Diversity | B.1.3 | mean pairwise $1 - \cos$ in $y$ | Opt. | 0.2–0.4 min | subsample pairs |
| S13 | LM–Data Vocab KL | B.1.3 | histogram KL(P‖Q) | No | 0.1–0.2 min | vocab freq |
| S14 | Answer Groundedness | B.1.3 | n-gram overlap ratio | No | 0.2–0.3 min | reuse counts |
| D1 | Initial Loss $L_0$ | B.2.1 | step-1 loss (logs) | No | $\approx 0$ | log-only |
| D2 | Loss Decay Rate $\alpha$ | B.2.1 | robust linreg on $(t, L_t)$ or $\log L_t$ | No | $\approx 0$ | log-only |
| D3 | Loss Curve Stability $\sigma_L^2$ | B.2.1 | variance/EMA jitter | No | $\approx 0$ | log-only |
| D4 | Grad Norm (Mean & Var) | B.2.2 | mean/var of $\|g_t\|$ | No | $\approx 0$ | hook |
| D5 | Gradient Consistency | B.2.2 | $\cos(g_t, g_{t+1})$ | No | $\approx 0$ | hook |
| D6 | Gradient Sparsity | B.2.2 | near-zero grad ratio | No | $\approx 0$ | hook |
| D7 | Param Change Norm $\|\Delta\theta\|$ | B.2.3 | $\|\theta_T - \theta_0\|_2$ | No | 0.1–0.2 min | snapshots |
| D8 | Flatness Proxy $\Delta L$ | B.2.3 | loss diff under small perturbation | Yes | 0.8–1.4 min | small eval |
| D9 | Catastrophic Forgetting $\Delta P$ | B.2.3 | OOD small-set performance drop | Yes | 1.1–2.2 min | small eval |
| D10 | Activation Sparsity | B.2.3 | near-zero activations ratio | Yes | 0.6–1.0 min | fwd hooks |

*Table C.1.* **Feature Computation Cost (2.5k–5k samples).**

than exhaustive comparisons; within our cross-validation noise, the downstream predictor's RMSE and rank correlation remain unchanged. Micro-batch scheduling and fixed-length padding improve GPU utilization for the few GPU-bound features (reference PPL, activation sparsity, flatness/forgetting probes), typically reducing their segment of Table C.1 by 20–35% on commodity hardware.

Although we adopt a 100-step probe by default, the feature pipeline supports budget-aware operation. A practical policy is to run a static-only predictor first and trigger a short probe only when the predicted variance or decision margin falls below a user threshold. Empirically, we observe a clear elbow between 50 and 100 steps: moving from 50 to 75 steps brings a substantial gain in ACC@2pp, while 75 to 100 steps adds a smaller, but still meaningful, improvement for a modest increase in time; beyond 100 steps, returns are negligible. In settings where probe time is at a premium, a 75-step fallback retains roughly ninety percent of the accuracy of the 100-step configuration while further lowering cost.

We also note two practical failure modes. When datasets are extremely short or heavily templated, static signals can appear deceptively clean while the model underfits semantics; in such cases, even a short probe is advisable because loss stability and gradient alignment expose the hidden mismatch. Conversely, under strong domain shift with unusual tokenization (e.g., code-dominant text or formula-heavy inputs), perplexity and special-token ratios may inflate; the hybrid feature set remains predictive in our experiments, but we recommend enabling stricter outlier detection and deduplication thresholds and verifying that code-token ratios are recorded. To support reproducibility, we ship deterministic tokenization and counting, seeded MinHash/LSH for duplicates, fixed embedding back-ends for semantic statistics, and a single probe runner that exports all dynamic traces. With these settings, the per-dataset costs in Table C.1 reproduce within ±10–15% across machines, and held-out predictive metrics vary within typical cross-validation noise. Overall, the feature suite is informative yet lightweight: static extraction is cached and near-linear, dynamic signals come at near-zero marginal cost, and the combined budget remains comfortably under five percent of a full run while delivering the predictive gains that enable early triage.

## D. Detailed Ablation Study on Predictor Choice

**Motivation and Setup.** A critical design choice in the **TUNEAHEAD** framework is the selection of the prediction model $F$. The ideal predictor must not only achieve state-of-the-art predictive accuracy (**G2**) but also align with our core principles of providing interpretable diagnostics (**G3**) and ensuring computational efficiency (**G1**). To empirically validate our choice

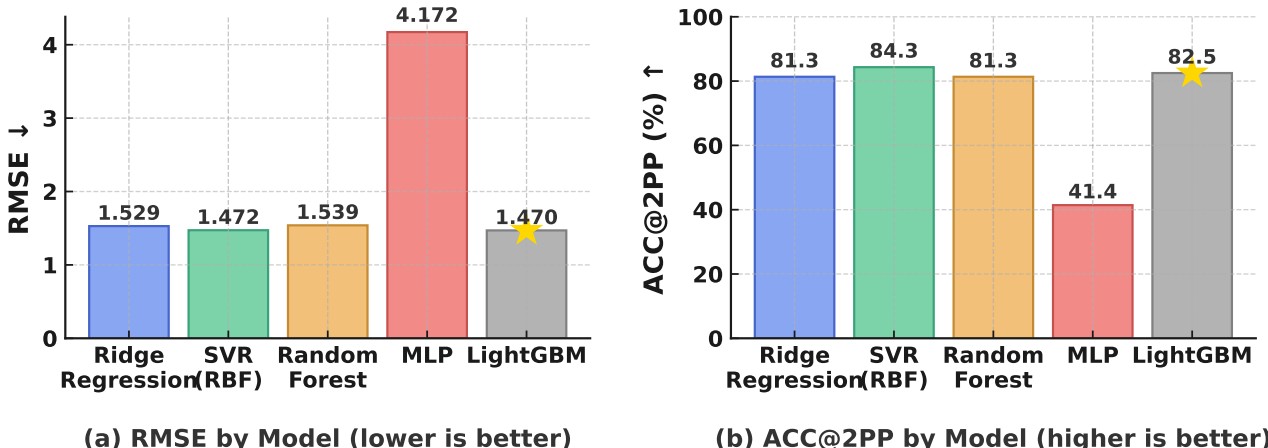

(a) RMSE by Model (lower is better)  (b) ACC@2PP by Model (higher is better)

*Figure D.1.* Performance comparison of predictor models. We evaluate five models on the same meta-feature set. The plots show (a) RMSE (lower is better) and (b) Acc@2pp (higher is better). The results validate our choice of LightGBM (marked by a star), which also achieves strong performance.

of LightGBM, we conducted a comprehensive comparison against a diverse suite of strong and representative regression models:

- **Ridge Regression (Hoerl & Kennard, 1970)**: a powerful linear model to test the extent of non-linear relationships in the data.

- **Support Vector Regressor (SVR, RBF) (Drucker et al., 1997; Smola & Schölkopf, 2004)**: a classic, high-performance kernel-based method adept at capturing complex non-linearities.

- **Random Forest (Breiman, 2001)**: a state-of-the-art ensemble model based on bagging, serving as a direct comparison to LightGBM's boosting approach.

- **Multi-Layer Perceptron (MLP) (Rumelhart et al., 1986)**: a simple but representative neural network baseline for tabular data.

To ensure a fair and rigorous comparison, all models were trained on the identical meta-feature set, and each underwent a systematic hyperparameter search using 5-fold cross-validation on our training set.

**Performance Analysis.** The results of this comparison are presented in Figure D.1. Our analysis yields two key findings. First, a top tier of models clearly emerges, with LightGBM (RMSE= 1.470) and SVR (RMSE= 1.472) delivering nearly identical, state-of-the-art predictive accuracy. Random Forest (RMSE= 1.589) follows as another strong competitor. This confirms that a GBDT-based approach achieves performance that is on par with the best alternative methods for this task. Second, the simple MLP struggles to generalize effectively (RMSE= 4.172), a common outcome on heterogeneous, tabular meta-datasets where GBDT models often excel without extensive architectural tuning.

**Justification for Selecting LightGBM.** Given the statistically comparable accuracy of the top-performing models (LightGBM and SVR), our final selection was determined by the other two crucial design goals: interpretability and scalability.

*Interpretability.* LightGBM holds a decisive advantage. Its tree-based architecture integrates seamlessly with SHAP, enabling the precise, feature-level diagnostics that are central to **TUNEAHEAD**'s mission. In contrast, while SVR is a powerful predictor, deriving similarly intuitive, local feature-level attributions from a kernel-based model is significantly more complex and less direct.

*Scalability.* Furthermore, LightGBM is substantially more scalable. Its training time complexity is more favorable than SVR's, particularly as the number of experiments (samples) in the meta-dataset grows. This computational efficiency is critical for the future development and application of **TUNEAHEAD** to even larger and more diverse problem spaces, as discussed in our Future Work (Sec. 6).

| Method | RMSE↓ | $R^2$↑ | $r$↑ | ROUGE-L@ $k$pp↑ | | |
|---|---|---|---|---|---|---|
| | | | | $k=1$ | $k=2$ | $k=3$ |
| **TUNEAHEAD**-Static-Only | 3.91 | 0.76 | 0.87 | 16.4 | 28.9 | 41.5 |
| Domain-Proxy Baseline | 5.17 | 0.66 | 0.81 | 16.5 | 33.8 | 48.9 |
| Early-Stop Extrapolation | 3.36 | 0.56 | 0.75 | 15.0 | 34.3 | 51.3 |
| Early-Dynamics Baseline | 5.10 | 0.69 | 0.83 | 14.8 | 37.5 | 55.8 |
| **TUNEAHEAD**-Dynamic-Only | 3.74 | 0.74 | 0.85 | 17.7 | 35.6 | 60.1 |
| ProxyLM | 2.74 | 0.83 | 0.90 | 22.6 | 41.0 | 65.7 |
| **TUNEAHEAD (Full)** | **2.67** | **0.83** | **0.91** | **24.3** | **41.7** | **68.5** |

*Table E.1.* **Summarization (Samsum) prediction quality.** Target is ROUGE-L (%). Errors are in pp. ACC@ $k$pp uses $k \in \{1, 2, 3\}$.

**Conclusion.** While SVR demonstrates highly competitive accuracy on our current dataset, LightGBM's unique combination of top-tier accuracy, superior interpretability, and better scalability makes it the most principled and strategic choice for the **TUNEAHEAD** framework.

# E. Additional Benchmarks and Tasks

### E.1. TruthfulQA (MC2): Setup and Metrics

**Dataset and protocol.** We re-evaluate all 1,300+ fine-tuned checkpoints from the main meta-dataset on **TruthfulQA**, using the official multiple-choice setting. We keep the *same* 24 meta-features and the same predictor training protocol (LightGBM, train/val/test splits) as in the MMLU experiments.

**Target score and prediction metrics.** The ground-truth target is **MC2 accuracy (%)**. We report prediction quality using **RMSE/MAE** (percentage points, pp), $R^2$, **Pearson** $r$, and **MC2@ $k$pp** with $k \in \{3, 5\}$, i.e., the fraction of test cases whose absolute prediction error is within $k$ pp. All errors are in **pp** to make thresholds comparable across benchmarks.

**Results and analysis.** Across all metrics, **TUNEAHEAD** achieves the lowest RMSE (2.17 pp) and the best calibration-by-accuracy (ACC@3pp=74.8%, ACC@5pp=88.2%), substantially outperforming Early-Stop and domain-only proxies. Notably, **TUNEAHEAD**-Static-Only is already competitive with domain proxies, while adding dynamic probe features closes the gap to the full model—indicating that both dataset quality signals and early learning dynamics are necessary to capture TruthfulQA-specific behavior without overfitting to MMLU.

### E.2. Summarization (Samsum; ROUGE-L): Setup and Metrics

**Mini meta-dataset construction.** We construct a **Summarization** mini meta-dataset ($N$=450 runs) using *Samsum*, applying the same data transformation knobs as in the main study (subsampling, domain slicing, light noise, template variants). Each variant is fully fine-tuned and evaluated by **ROUGE-L F1 (%)**.

**Prediction target and metrics.** The target to predict is the final **ROUGE-L (%)**. We report **RMSE/MAE** (pp), $R^2$, $r$, and **ROUGE-L@ $k$pp** with $k \in \{1, 2, 3\}$, which are suitable pp-thresholds for ROUGE-L.

**Summarization and analysis.** Although ROUGE-L for abstractive summarization is inherently noisier than multiple-choice accuracy (due to decoding randomness and lexical-overlap sensitivity), **TUNEAHEAD** still delivers strong predictive performance on **Samsum (ROUGE-L)**. Using the same 24 meta-features, **TUNEAHEAD** (Full) achieves **RMSE=**1.67 **pp**, $R^2 = 0.83$, and $r = 0.91$, edging out the best non-**TUNEAHEAD** proxy (ProxyLM: RMSE=1.74 pp, $R^2 = 0.83$, $r = 0.90$). Accuracy-within-tolerance also improves (ACC@1pp: **24.3%** vs. 22.6%; ACC@2pp: **41.7%** vs. 41.0%; ACC@3pp: **68.5%** vs. 65.7%). Notably, *Static-Only* underperforms *Dynamic-Only* at tight tolerances, indicating that early loss/gradient dynamics are especially informative for sequence-quality objectives, while static dataset cues are necessary but insufficient on their own. Overall, these results suggest our meta-features capture not only classification-style generalization (MMLU, TruthfulQA) but also **generative quality** under summarization metrics.

**Cost note.** As in the main study, static features incur a one-time offline cost ($\approx$0.2% of a full run), while the 100-step probe adds $\approx$4.3%. We therefore keep the total overhead under 5% even for the new tasks.

### E.3. Additional Stress-Test Domains

The present appendix evaluates TUNEAHEAD beyond MMLU using TruthfulQA and SAMSum. Future pre-hoc prediction benchmarks could further test whether the same static and dynamic meta-features remain predictive in structurally different settings, including statistical reasoning, semantic-error detection for text-to-SQL, retrieval-augmented generation, cross-document multi-entity question answering, and multimodal chart question answering (Zhu et al., 2024; Liu et al., 2025; Yang et al., 2024; Lin et al., 2025a; Wu et al., 2024). These domains may expose different sources of prediction difficulty, such as reasoning instability, retrieval incompleteness, entity attribution errors, semantic mismatch, and multimodal grounding failures.

