# OpenReview forum: "TuneAhead: Predicting Fine-tuning Performance Before Training Begins"
_ICML.cc/2026/Conference — ICML 2026 regular_

### Official Review · Reviewer_4nJS · 2026-03-03

**Soundness:** 2
**Presentation:** 3
**Significance:** 2
**Originality:** 2
**Overall Recommendation:** 3
**Confidence:** 4

**Summary:**

This paper introduces TuneAhead, a meta-learning framework for predicting LLM fine-tuning performance before full training is executed. The authors formulate the problem as learning a mapping from a compact meta-feature vector to expected downstream performance, where the vector combines 14 static dataset descriptors (lexical diversity, semantic properties, reference perplexity, etc.) with 10 dynamic features extracted from a short 100-step probe run (loss dynamics, gradient statistics, landscape flatness). A LightGBM regressor serves as the predictor, and SHAP-based attribution produces interpretable diagnostics identifying which dataset or optimization properties drive predicted outcomes. Evaluated across 1,300+ fine-tuning runs on Qwen2.5-7B-Instruct (MMLU), with additional cross-model and cross-task experiments, TuneAhead outperforms baselines including ProxyLM and Early-Stop Extrapolation, and is shown to enable substantial compute savings by flagging likely-failure runs before training begins.

**Compliance With Llm Reviewing Policy:**

Affirmed.

**Final Justification:**

I will keep my original decision (weak reject), since this work focuses on an interesting problem, but their results raise questions about the efficiency and robustness of the framework they claim. In their rebuttal, they did not fully resolve my concerns, so in this case, my main concerns remain, and I keep the original decision.

**Key Questions For Authors:**

Following the weaknesses, I have the following questions:

1. Was the SHAP-guided feature selection in Appendix A.3 performed exclusively on training/validation data, with zero contact with the held-out test set? If threshold values, percentile cutoffs, or feature retention decisions were informed by test-set performance in any way, the reported metrics are optimistic. Please provide a precise description of which data partitions were used at each stage of the selection pipeline.

2. For a fixed dataset Di evaluated at two different learning rates (e.g., 1e-5 and 3e-5), how much do the SHAP attributions of dynamic features shift? If loss stability and gradient norms primarily reflect the hyperparameter choice rather than intrinsic dataset properties, the claim that SHAP diagnostics guide "targeted data-centric improvements" would require qualification.

3. Rather than training a fresh predictor per model family, did you evaluate a predictor trained on Qwen2.5-7B applied directly to Llama-3-8B without retraining? Even if performance degrades, understanding which features transfer and which do not would provide actionable insight into the model-specificity of dynamic versus static features.

4. What is the exact numerical threshold used to define "success" in the binary classification results (89.4%/91.0% recall figures)? How sensitive are these figures to the threshold value? Is the threshold set on the training/validation split and held fixed for the test set, or is it tuned on the test set?

**Limitations:**

yes

**Strengths And Weaknesses:**

Strengths:

1. Principled hybrid feature design: The decomposition of meta-features into static (dataset-intrinsic) and dynamic (model-interaction) categories is well-motivated. Static features capture dataset quality prior to any training, while dynamic probe features expose model-specific learnability that is invisible to static analysis alone. This complementarity is not merely claimed but rigorously validated through the four-bucket ablation in Table 4, which shows the full model rescuing cases where either branch individually fails.
2. Strong empirical results with thorough statistical rigor: The evaluation covers 1,300+ runs with three-seed averaging, bootstrap confidence intervals (N=1,000 resamples), and paired permutation tests. TuneAhead cuts RMSE by 30% relative to the strongest baseline (ProxyLM) and achieves 95.1% of predictions within ±3pp, which is the tolerance regime practitioners care about for go/no-go decisions. These gains are consistent across MMLU, TruthfulQA, and SAMSum, across three model families, and across all seven per-domain MMLU breakdowns (Table A.1).
3. Actionable diagnostic loop: The paper goes beyond prediction to demonstrate actionability. The failure case study in Section 5 (Exp-2) traces a predicted failure through SHAP attributions to specific prescriptions (lower learning rate, semantic deduplication, outlier removal), then validates those interventions empirically - improving final MMLU score from 20.2% to 48.7%. This end-to-end demonstration is a genuine contribution over proxy-score baselines that return only an opaque scalar.
4. Data-driven feature selection: Rather than hand-picking features, the authors start from 50+ candidates and apply a SHAP-guided selection pipeline with three principled criteria (global importance, directional consistency, redundancy pruning). The resulting 24-feature set is compact and interpretable, and the selection procedure is documented in sufficient detail in Appendix A.3 to be reproducible.
5. Calibration analysis: Figure A.1 shows that TuneAhead's predictions are not only accurate in rank correlation but also well-calibrated in absolute probability, remaining close to the diagonal across the full predicted score range. This property is critical for threshold-based go/no-go decisions and distinguishes TuneAhead from baselines that systematically over- or underestimate success probability.

Weaknesses:

1. Potential leakage in feature selection: The SHAP-guided feature selection procedure (Appendix A.3) trains a preliminary LightGBM on the candidate pool and prunes based on SHAP importance. The paper does not explicitly confirm that this selection was performed exclusively on training/validation folds, with zero access to the held-out test set. If any test data influenced which features were retained, the reported test metrics are optimistic. This is a critical methodological concern that must be clarified.
2. Hyperparameter-probe confound in dynamic features: Dynamic features such as loss decay rate, gradient norm, and gradient consistency are extracted from a 100-step probe at a specific (Di, Hj) configuration. For a fixed dataset Di evaluated under two different learning rates Hj and Hj', these features will differ substantially. The SHAP attribution "gradient instability" may therefore reflect a poorly chosen learning rate rather than an intrinsic dataset defect, undermining the diagnostic interpretability claim. A small experiment holding the dataset fixed and varying only the learning rate would clarify whether dynamic features are primarily dataset-descriptive or hyperparameter-descriptive.
3. Narrow hyperparameter sweep: The LoRA search covers only 2 learning rates and 3 batch sizes (6 combinations), which is narrow relative to the diversity practitioners face. This means many of the 1,300+ runs share identical hyperparameter settings, potentially inflating apparent generalization. It is unclear whether TuneAhead remains predictive when LoRA rank, dropout, number of epochs, or warmup schedules are also varied.
4. Compute savings figures are inconsistent and context-dependent: The abstract reports 58.4% compute savings, Figure 2 illustrates 37.4% for a worked example, and the two figures are not reconciled in the main text. The savings depend critically on the assumed base failure rate in the practitioner's pool, the predictor's precision and recall at the chosen threshold, and the probe overhead. A formula or sensitivity plot showing how savings vary with these inputs would make the claim much more useful and honest.
5. Success/failure threshold is underspecified: The binary classification metrics (89.4% recall on successes, 91.0% on failures) depend on a performance threshold defining "success," but this threshold is not explicitly specified in the main paper. The choice of threshold is consequential: a tight threshold produces few positives and makes precision/recall highly sensitive to small shifts in predicted score. The paper should specify how this threshold is chosen and report sensitivity to its value.
6. Cross-model generalization experiments are underpowered: The Llama-3-8B and Qwen2-0.5B meta-datasets contain only ~400 and ~450 runs, respectively, compared to 1,300+ for Qwen2.5-7B. This size disparity disadvantages all methods equally, making it difficult to draw strong conclusions about cross-architecture generalization. More importantly, the experiments retrain a fresh predictor per model family rather than testing zero-shot or few-shot transfer from one family to another, which is the practically relevant generalization scenario.
7. No convergence or approximation error analysis: The framework involves multiple approximations (SHAP-guided feature pruning, LightGBM as surrogate, z-score normalization). There is no analysis of how errors in individual components compound, or under what conditions the meta-predictor is expected to degrade. A theoretical or empirical sensitivity analysis would strengthen the foundation.
8. Generalization to PEFT methods beyond LoRA is untested: All experiments use LoRA fine-tuning. It is unclear whether the dynamic probe features - which depend on gradient statistics from LoRA adapter parameters - remain informative for full-parameter fine-tuning, QLoRA, or other PEFT variants. This limits the scope of the contribution.

---

> ### Author Rebuttal · Authors · 2026-03-31
>
> We thank the reviewer for the detailed and careful reading. Because the rebuttal has **5000 characters** limit, we focus here on the reviewer’s **four specific questions**.
>
> **Q1. Was SHAP-guided feature selection performed strictly without touching the held-out test set?**
> **Yes.** The SHAP-guided feature selection in Appendix A.3 was performed **exclusively on the training/validation data**, with **zero contact** with the held-out test set. The meta-dataset was split into **72% train/validation** and **28% held-out test**; all stages of SHAP-guided selection (preliminary LightGBM fitting, SHAP computation, and pruning/CV checks) were performed **only within the 72% train/validation portion**. No percentile cutoffs, thresholds, or feature-retention decisions were ever informed by the test set; the test set was used **only once** for final evaluation of the frozen 24-feature model. Because of the reviewer’s concern, we re-checked the code and reran the framework under this protocol; the results remain essentially unchanged.
>
> **Q2. Are the dynamic features mainly describing the dataset, or just the chosen hyperparameters?**
> For a fixed dataset \(D_i\), the dynamic features do shift when the learning rate changes (e.g., from \(1\times10^{-5}\) to \(3\times10^{-5}\)). This is expected and by design: the 100-step probe is run with the specific hyperparameter configuration \(H_j\), so the dynamic features capture the **joint interaction** between the dataset and the chosen hyperparameters. Hyperparameters are therefore part of the input.
>
> This does **not** undermine the diagnostic value of SHAP. Rather, it allows the attribution to identify both **data-related issues** and **hyperparameter-related issues**. A concrete example already appears in Section 5.2 / Figure 5(b): for one failure case, SHAP highlighted both **extremely low loss stability / poor landscape flatness** and **high duplicate rate + embedding outliers**. Following these attributions, we **simultaneously** lowered the learning rate and performed **semantic de-duplication + outlier removal**, improving the final MMLU score from **20.2% to 48.7%**. We will clarify this joint nature in Section 5.2 and the discussion.
>
> **Q3. Did we test direct predictor transfer across model families?**
> We agree that **zero-shot cross-family transfer degrades substantially**, so the paper should not be interpreted as claiming universal predictor transfer. However, preliminary **within-family** experiments suggest a practical path forward. A zero-shot predictor trained on **Qwen2.5-7B** performs poorly when applied directly to **Qwen2-0.5B**, but after only **70 calibration runs**, the calibrated predictor achieves **RMSE = 2.43** and **Acc@3pp = 82.0**, already outperforming the predictor trained from scratch on the **450-run Qwen2-0.5B** meta-dataset reported in the paper. We therefore view **family-level transfer with lightweight calibration** as the more realistic deployment path, and we will clarify this scope more explicitly in the final version.
>
> **Q4. What is the exact success threshold, was it tuned on test, and how should the compute-saving numbers be interpreted?**
> The exact threshold is **55%**, and it was a **pre-set value**, not tuned on the test set. We use this threshold in practice as a simple go/no-go target when training some smaller models, which is why we adopted it here.
>
> Most importantly, this threshold has **zero effect on TuneAhead’s regression accuracy itself**. TuneAhead predicts continuous final performance; the threshold is applied **only afterwards** to convert those regression outputs into a binary screening policy. It therefore affects only deployment quantities (**saving / success retention / failure filtering**), **not** RMSE / \(R^2\) / Pearson \(r\) / Acc@k. **Due to space, we refer to our response to Reviewer U5rR for the full threshold-sensitivity table**.
>
> This also explains the reviewer’s concern about the **58.4%** vs **37.4%** compute-saving figures. These correspond to **different contexts**, not inconsistent estimates of one universal quantity: **58.4%** is the aggregate saving under the held-out evaluation protocol at the chosen operating point, whereas **37.4%** in Figure 2 is an **illustrative worked example** on a small candidate pool (10 runs). We agree that the text should reconcile this more clearly, and we will revise the final version to make explicit that compute savings are inherently **context-dependent**, driven by the candidate-pool success/failure rate, the predictor’s screening behavior at the chosen threshold, and the probe overhead.
>
> We thank the reviewer again. We will scope more carefully the claims on cross-family transfer and beyond **LoRA-based PEFT**. In particular, the generalization of the probe features to **QLoRA**, other PEFT variants, or full-parameter fine-tuning is not yet established, and we will state this limitation more explicitly in the final version and **Future Work**.

---

> > ### Author Rebuttal · Reviewer_4nJS · 2026-04-02
> >
> > Thank you for the rebuttal. The authors have addressed several of my concerns. After carefully reviewing the rebuttal, I remain inclined to maintain my score.

---

> > > ### Author Response · Authors · 2026-04-07
> > >
> > > Thank you for the follow-up. Beyond the four protocol questions addressed in the rebuttal, we also want to clarify four remaining scope issues from your original review:
> > >
> > > **(W3) Narrow hyperparameter sweep.**
> > > The current benchmark is built on an intentionally **controlled LoRA search space**, not an exhaustive practitioner-scale sweep over rank, dropout, epochs, or warmup schedules. Accordingly, the paper supports conclusions within a realistic but still limited hyperparameter regime; it does **not** claim coverage of the full practical search space.
> > >
> > > **(W6) Cross-model generalization experiments are underpowered.**
> > > We agree that **cross-family generalization remains a real difficulty** in the current version, and the auxiliary Llama-3-8B / Qwen2-0.5B results should be read primarily as evidence of **framework portability**, not as strong evidence of universal transfer across model families. At the same time, our more recent calibration results suggest that **within-family or closely related-architecture transfer** can be made practical with only a small number of additional calibration runs. We therefore view **family-level transfer with lightweight calibration** as the more realistic deployment path at this stage, rather than zero-shot transfer across arbitrary families.
> > >
> > > **(W7) No convergence or approximation-error analysis.**
> > > The current paper is an empirical framework paper and does **not** provide a full theoretical analysis of how approximation errors from SHAP-guided pruning, LightGBM surrogacy, and normalization compound. This is a genuine theoretical limitation of the present version, but it does not affect the empirical protocol clarifications given in the rebuttal.
> > >
> > > **(W8) Generalization beyond LoRA-based PEFT is untested.**
> > > All experiments in the current submission are based on **LoRA-style PEFT**. The paper therefore does **not** establish that the same probe features remain equally informative for **QLoRA**, other PEFT variants, or **full-parameter fine-tuning**. This is indeed a limitation of the current paper, and also a natural direction for **Future Work**: one paper cannot realistically cover all transfer settings and PEFT variants at once, and we will state this boundary more explicitly in the final version.
> > >
> > > These points narrow the scope of the current submission, but they do not change the main empirical conclusion supported by the paper: within the studied LoRA-based setting, static + short-run dynamic signals provide a stable and useful basis for pre-screening and diagnosis. We will state these boundaries more explicitly in the final version and include them in **Future Work**.

---

### Official Review · Reviewer_U5rR · 2026-03-09

**Soundness:** 3
**Presentation:** 4
**Significance:** 4
**Originality:** 4
**Overall Recommendation:** 5
**Confidence:** 5

**Summary:**

This paper explores whether the outcome of a fine-tuning run can be predicted prior to performing full training. To answer this, the authors propose TuneAhead, which represents each fine-tuning run as a meta-feature vector composed of two parts: static dataset features (e.g., data size, diversity, perplexity) and dynamic probe features extracted from a short 100-step probe run (e.g., loss, gradient statistics, flatness-related signals). A LightGBM predictor is then used to estimate final performance, while TreeSHAP is used to provide interpretable diagnostics. The paper makes three main contributions. First, it formulates fine-tuning success prediction as a standalone problem, rather than treating it as a side product of training-curve extrapolation or proxy scoring. Second, it proposes a simple and coherent framework that combines static + dynamic low-cost signals and supports interpretable diagnosis through SHAP. Third, it builds a meta-dataset with 1300+ fine-tuning runs and evaluates the approach on a main setting with Qwen2.5-7B-Instruct, as well as additional settings including Llama-3-8B-Instruct, Qwen2-0.5B, TruthfulQA, and SAMSum.

**Compliance With Llm Reviewing Policy:**

Affirmed.

**Final Justification:**

The rebuttal addressed my main concerns and clarified the scope and practical interpretation of the paper. In particular, the authors appropriately refined the framing of the probe stage, acknowledged the limits of cross-family generalization, and made the threshold-dependent deployment trade-off more explicit. I find these clarifications sufficient, and I believe my original score already fairly reflects the paper’s contribution, so I will keep it as 5.

**Key Questions For Authors:**

**Q1. Why is a 100-step probe the right choice, and how stable is this choice across settings?**
The paper provides an elbow-style analysis for 100 steps, which is useful. However, I would like to know whether this conclusion still holds for larger models, longer-context tasks, or full-parameter fine-tuning. Is 100 steps a generally good default, or is it mainly specific to the current setup?

**Q2. How sensitive are the results to the success threshold?**
Since the compute-saving claim depends on converting predicted performance into a success/failure decision, it would be helpful to see a more detailed analysis of threshold sensitivity. For example, how much do the savings and error trade-offs change under different thresholds?

**Q3. Do the authors have results on larger models?**
The current experiments are useful, but they are still relatively limited in scale. It would strengthen the paper if the authors could include evidence on larger model settings, or at least discuss more clearly how the method is expected to behave there.

**Limitations:**

Yes

**Strengths And Weaknesses:**

***Strengths***

**1. Strong and well-motivated problem setting.**
The paper asks a very practical question: not “how to fine-tune better,” but whether a run is worth starting at all. This is highly relevant to both research and real-world deployment, where fine-tuning is costly and trial-and-error is common.

**2. The method is simple, reasonable, and well-structured.**
The combination of static and dynamic signals makes intuitive sense. Static features capture dataset-level properties, while dynamic features capture the early interaction between the model and the data. These two views are naturally complementary. I also appreciate that the paper does not simply throw in many features, but performs SHAP-guided feature selection to reduce the candidate pool from 50+ features to 24.

**3. Clear writing and overall good organization.**
The paper is generally easy to follow. The motivation, problem definition, two-stage pipeline, baselines, metrics, and appendix details are presented in a fairly organized way. The overall story is coherent.

***Weaknesses***

**1. The phrase “before training begins” is somewhat too strong.**
Strictly speaking, this is not a fully training-free prediction method, since it still requires a standardized 100-step probe run and records training-related signals such as loss and gradients. Although the overhead is small, the title and framing may give the impression that no training is needed at all. I think the presentation should be more precise here.

**2. Generalization is still a major limitation, especially across model families.**
The paper itself acknowledges that the predictor is still fairly tied to the model family. Since the dynamic features depend on optimization behavior, transferring the predictor across architectures or training regimes may require retraining. In other words, the paper shows strong within-family prediction, but it is still far from a truly general fine-tuning predictor.

**3. The compute-saving claim depends on threshold choice and deployment policy.**
The reported compute savings are practically interesting, but they rely on defining success/failure using a threshold. In practice, the choice of threshold matters a lot, and the cost of mistakenly filtering out a potentially strong run may also be high. The paper would be stronger if it discussed this trade-off in more depth.

---

> ### Author Rebuttal · Authors · 2026-03-31
>
> We sincerely thank the reviewer for the careful reading, thoughtful comments, and strong overall support. We agree that the three concerns raised by the reviewer are important for making the paper more precise and more useful in practice.
>
> **(1) On the wording “before training begins” and on the 100-step probe (Weakness 1; Q1).**
> The reviewer is absolutely right that the phrase **“before training begins”** is too strong in its current form. TuneAhead is not training-free in the literal sense, since it relies on a standardized short probe run. What we intended to mean is more precisely **before full training begins / before committing the full fine-tuning**, and we will revise the final version to make this wording much more explicit.
>
> We also agree that the current **100-step probe** should not be interpreted as a universal constant. In the present paper, **100 steps** is only a **strong empirical default for the current PEFT setting**, supported by an elbow in the cost–accuracy trade-off, but we do **not** claim that it will remain optimal for **larger models, longer-context tasks, or full-parameter fine-tuning**. In that sense, the current choice remains partly empirical. We fully accept the reviewer’s suggestion that an **adaptive stopping criterion** would be a more principled solution, and we will add adaptive probe selection more explicitly to the **Future Work** section in the final version.
>
> **(2) On generalization and the reviewer’s concern that TuneAhead is still far from a universal predictor (Weakness 2).**
> We agree with the reviewer that **cross-family generalization remains a real difficulty**, and the paper should not be read as claiming a universal fine-tuning predictor. We will revise the final version to make this limitation more explicit.
>
> At the same time, we believe a more practical intermediate direction is **within-family reuse with lightweight calibration**. Following this idea, we conducted a preliminary additional experiment: a **zero-shot** predictor trained on **Qwen2.5-7B** performs poorly when applied directly within-family, but after only **70 additional calibration runs**, the calibrated predictor achieves **RMSE = 2.43** and **Acc@3pp = 82.0** on **Qwen2-0.5B**, already outperforming the predictor trained from scratch on the **450-run Qwen2-0.5B** meta-dataset reported in the paper. This suggests that, while naive zero-shot transfer is insufficient, **lightweight intra-family calibration** may be a promising practical solution. We will add this discussion more clearly in the final version while keeping the claim appropriately scoped.
>
> **(3) On threshold sensitivity and the interpretation of compute savings (Weakness 3; Q2).**
> We fully agree with the reviewer that the reported **compute savings are threshold- and policy-dependent**. Importantly, the threshold choice has **no effect on TuneAhead’s regression accuracy itself**; it only changes how the continuous predictions are converted into a go/no-go decision, and therefore changes the trade-off between saving compute and retaining successful runs.
>
> To make this clearer, we will add a threshold-sensitivity discussion such as the following:
>
> | Success threshold | True success rate in metadataset | Runs sent to full fine-tuning | Net compute saving | Successful runs retained | Failure filtering rate (TNR) |
> |---|---:|---:|---:|---:|---:|
> | 50% | 44.0% | 43.4% | 51.6% | 96.0% | 95.5% |
> | 55% | 38.2% | 36.1% | 58.9% | 94.5% | 96.0% |
> | 60% | 30.3% | 27.6% | 67.4% | 91.3% | 94.8% |
> | 65% | 19.1% | 17.7% | 77.3% | 88.0% | 93.3% |
> | 70% | 2.9% | 8.8% | 86.2% | 84.0% | 92.5% |
>
> This makes the deployment trade-off more explicit: stricter thresholds save more compute, but also retain fewer successful runs. We agree that this discussion should be clearer in the paper, and we will strengthen the final version accordingly.
>
> **(4) On larger models (Q3).**
> We agree that the current paper does not yet provide sufficiently strong evidence beyond the scale. We are already exploring whether the predictor can be reused on a larger same-family model (**Qwen2.5-14B**). At the moment, the rebuttal-time evidence is still too limited for a strong claim, but we want to be transparent about what we currently observe. Using a **Qwen2.5-7B-based predictor** with **50 calibration runs**, we evaluated **42 cases** on **Qwen2.5-14B**:
>
> | Metric on Qwen2.5-14B (42 cases) | Correct predictions |
> |---|---:|
> | **Acc@2pp** | **23 / 42** |
> | **Acc@3pp** | **31 / 42** |
>
> We view these results as encouraging but still preliminary, and not yet strong enough to claim convincing large-scale validation. We therefore present them only as ongoing exploration, and we will scope the final version accordingly.
>
> Again, we sincerely thank the reviewer for the careful reading and highly constructive feedback. We believe these clarifications will make the paper more precise and better aligned with its intended scope, and we will incorporate them in the final version.

---

> > ### Author Rebuttal · Reviewer_U5rR · 2026-04-01
> >
> > Thank you for the detailed rebuttal. My concerns have been addressed, and I think my original score is already fair. I will keep my score.

---

> > > ### Author Response · Authors · 2026-04-07
> > >
> > > Thank you very much for your follow-up and for taking the time to read our rebuttal carefully. We truly appreciate your thoughtful review and are glad that our response addressed your concerns. We also appreciate your fair and supportive assessment of the paper. We will make sure to incorporate these clarifications into the final version.

---

### Official Review · Reviewer_7YGK · 2026-03-12

**Soundness:** 3
**Presentation:** 3
**Significance:** 4
**Originality:** 3
**Overall Recommendation:** 5
**Confidence:** 3

**Summary:**

This paper proposes TuneAhead, a lightweight framework for predicting the outcome of an LLM fine-tuning run prior to full training. The method encodes each run using static dataset features and dynamic signals extracted from a fixed 100-step probe run, mapping them to performance predictions via LightGBM with SHAP-based feature attribution. Evaluated on over 1,300 fine-tuning runs, TuneAhead consistently outperforms prior baselines, demonstrating that static and dynamic features are highly complementary for reliable pre-screening.

**Compliance With Llm Reviewing Policy:**

Affirmed.

**Final Justification:**

I appreciate the authors' response. I find the motivation and content of this paper to be excellent. The authors' candid responses further enhance the practical utility of the work. Consequently, I have raised my score to 5. I recommend that the authors incorporate these findings into the final version. In my view, discussing the limitations of the method is not a weakness, but rather a responsible scholarly practice.

**Key Questions For Authors:**

See weaknesses.

**Limitations:**

yes

**Strengths And Weaknesses:**

## Strengths
1. The paper addresses a practically important problem for LLM post-training, predicting whether an SFT run is worth the compute budget before full training.

2. The proposed framework is simple and well-motivated, combining static dataset features and short-run dynamic signals in a way that is both effective and interpretable.

3. The evaluation is fairly extensive, with strong baseline comparisons, useful ablations, and clear evidence that static and dynamic features provide complementary information.

## Weaknesses

1. Lack of an adaptive probe stopping criterion. The fixed 100-step probe is essentially a static heuristic, which may waste compute on simple tasks or under-extract features on complex ones. Introducing an adaptive stopping criterion (e.g., dynamically halting based on predictive uncertainty, decision margins, or when dynamic statistics stabilize) would elevate the method from a simple rule of thumb to a rigorous budget-control mechanism.

2. Unverified intra-family model generalization. While the paper notes the difficulty of cross-family generalization, it overlooks the highly feasible *intra-family* transfer (e.g., from Qwen2.5-7B to 1.5B). Since models within the same family share architectural designs and pre-training distributions, their early training dynamics are likely highly correlated. Verifying this intra-family transferability would drastically reduce the high cost of recollecting meta-datasets, enhancing the framework's practical utility.

---

> ### Author Rebuttal · Authors · 2026-03-31
>
> We sincerely thank the reviewer for the careful reading and highly insightful suggestions. We especially appreciate that the reviewer clearly understood the intended role of TuneAhead and identified two directions that are both practically important and technically well-motivated. We agree that these comments exposed two genuine limitations of the current version, and they have been extremely helpful for sharpening both the scope and the practical deployment story of the method.
>
> **On the fixed 100-step probe.**  (W1)
>
> -We agree with the reviewer that this is a limitation of the current version. Although our paper shows that **100 steps** is a strong empirical trade-off point in the current setup, with a clear elbow in the cost–accuracy curve and diminishing returns beyond 100 steps, we do **not** view it as a universal value for **larger models, more complex tasks, or other fine-tuning regimes**. In that sense, the current choice remains somewhat empirical.
>
> -We fully agree that introducing an **adaptive stopping criterion** would make the framework more principled and more useful in practice. We will therefore revise the final version to make this limitation clearer and to add **adaptive stopping based on uncertainty / decision margin / stabilization of probe statistics** as a more explicit direction in **Future Work**.
>
> **On intra-family transfer.**  (W2)
>
> -We also think the reviewer identified a genuine weakness of the current paper. This is indeed a more feasible and practically meaningful setting than broad cross-family transfer, and we agree that we should have discussed it much more clearly in the original submission.
>
> -The current paper mainly validates **framework portability** across settings, but that is not the same as directly testing whether a predictor can be reused efficiently within a family. Following the reviewer’s suggestion, we conducted a preliminary additional experiment:
>
> -A **zero-shot** predictor trained on **Qwen2.5-7B** performs poorly when applied directly within-family(Qwen 2-0.5B and Qwen 2.5-14B).
>
> -However, after only **70 additional calibration runs**, the calibrated predictor achieves **RMSE = 2.43** and **Acc@3pp = 82.0** on **Qwen2-0.5B**, already outperforming the predictor trained from scratch on the **450-run Qwen2-0.5B** meta-dataset reported in the paper, where **RMSE = 3.75** and **Acc@3pp = 74.6**  . This suggests that, while naive zero-shot transfer is insufficient, **lightweight intra-family calibration** may substantially reduce the cost of recollecting a full target-family meta-dataset.
>
> At the same time, we want to be transparent about the current scope of this new evidence. So far, we have only tested this within the **Qwen family**, and we are still exploring what calibration budget is most efficient and stable. We also have not yet carried out sufficient experiments on larger same-family models. We will therefore present this as encouraging preliminary evidence rather than as a finalized claim, and we will strengthen the final version accordingly.
>
> Overall, we appreciate the reviewer’s framing very much. We believe the current paper establishes that low-cost pre-screening with static + short-run dynamic signals is already effective and interpretable, and the reviewer’s two suggestions point directly to the most valuable next steps. We will incorporate both clarifications in the final version.

---

> > ### Author Rebuttal · Reviewer_7YGK · 2026-04-01
> >
> > I appreciate the authors' response. I find the motivation and content of this paper to be excellent. The authors' candid responses further enhance the practical utility of the work. Consequently, I have raised my score to 5. I recommend that the authors incorporate these findings into the final version. In my view, discussing the limitations of the method is not a weakness, but rather a responsible scholarly practice.

---

> > > ### Author Response · Authors · 2026-04-07
> > >
> > > Thank you very much for your generous follow-up and for raising your score to 5. We sincerely appreciate your careful reading, thoughtful suggestions, and encouraging assessment of the paper’s motivation and practical value. We are especially grateful for your view that openly discussing limitations is a strength rather than a weakness; this was very meaningful to us. We will incorporate the additional findings and the clarified discussion of limitations into the final version.

---

### Official Review · Reviewer_wy6n · 2026-03-12

**Soundness:** 2
**Presentation:** 3
**Significance:** 3
**Originality:** 2
**Overall Recommendation:** 4
**Confidence:** 3

**Summary:**

This paper proposes TuneAhead, a method for predicting LLM fine-tuning performance before full training begins. The authors first integrate static and dynamic characteristics into a unified meta feature vector. Then, a LightGBM model is utilized to assess the ultimate performance, with SHAP attribution employed to elucidate the outcomes. The experimental procedures were executed utilizing the MMLU and TruthfulQA datasets, incorporating the Qwen and Llama models.

**Compliance With Llm Reviewing Policy:**

Affirmed.

**Final Justification:**

Thank you for the detailed response. I believe the paper's findings will be of great interest to the LLM research community, and I will maintain my original positive score.

**Key Questions For Authors:**

1. Could the authors provide a full computational cost comparison between TuneAhead and other baseline methods? I ask because TuneAhead is not training-free; it still requires training a model on feature vectors.
2. Could the authors explain how hyperparameters are adjusted when constructing the dataset and training the predictors? Does switching to another model or dataset require significant effort in parameter tuning?
3. Could TuneAhead be useful for efficient data selection in LLM fine-tuning or instruction tuning?

**Limitations:**

yes

**Strengths And Weaknesses:**

- Soundness: The submission is technically sound. Both the method design and evaluation are reasonable, and the experimental results strongly indicate that TuneAhead has better predictive ability for forecasting fine-tuning performance. The SHAP-guided selection can also help diagnose failure cases.

- Presentation: The presentation is clear, and the process for creating both static and dynamic features is easy to follow. In addition, the appendix provides more results, analyses, and computational costs.


- Significance:  Fine-tuning LLMs is very common in real-world applications today. The authors propose a method for predicting final performance before full training begins, which could be useful for many researchers in the field of large language models.

- Originality: The use of both static and dynamic features to predict fine-tuning performance is a new. However, it remains unclear whether the predictor has strong cross-model or cross-dataset generalizability. I.e., whether a predictor trained on one model and dataset can be applied to predict fine-tuning performance for other models.

---

> ### Author Rebuttal · Authors · 2026-03-30
>
> We sincerely thank the reviewer for the careful reading, positive assessment, and constructive suggestions. We also appreciate the recognition that TuneAhead achieves stronger predictive performance than prior baselines and that its SHAP-based analysis can help diagnose failure cases. Below we address the three questions and clarify how we will strengthen the final version.
>
> **(1) Full computational cost comparison.**
> We agree that the paper should make the cost story clearer. TuneAhead is **not training-free in the strict sense**, since the LightGBM predictor is fit offline on the meta-dataset. However, this offline fitting step is **essentially negligible in practice**: the predictor is a lightweight tabular regressor trained on only about **1,300 meta-examples**, each fit takes **less than 40 seconds** in our runs, and it requires **no GPU at all**. In practice, the real cost comes from obtaining probe signals for candidate runs, not from fitting the predictor itself.
>
> The practically relevant comparison is therefore the **deployment-time screening cost per candidate run**, shown below:
>
> | Method | Deployment runtime per candidate run (min) | Deployment cost (% of one full fine-tuning run) |
> |---|---:|---:|
> | Domain-Proxy Baseline | 0.7–1.5 | 0.4–0.8% |
> | ProxyLM | 4.5–9.0 | 2.5–5.0% |
> | Early-Stop Extrapolation | 7.5–7.8 | 4.2–4.3% |
> | Early-Dynamics Baseline | 7.8–8.0 | 4.3–4.4% |
> | TuneAhead (Full) | 8.1–8.3 | 4.5–4.6% |
>
> As shown above, a full TuneAhead prediction costs only **8.1–8.3 minutes per candidate run**, or about **4.5–4.6%** of one full fine-tuning run. This is only slightly above other probe-based baselines, while providing substantially stronger predictive accuracy and diagnostic value. Domain-Proxy is cheaper but much weaker in predictive quality. ProxyLM is reported as a range because its runtime depends on the chosen proxy family, proxy size, and the number of proxy evaluations.
>
> In other words, compared with other probe-based baselines, TuneAhead adds only a **small marginal overhead** on top of the same short probe, while yielding a much larger gain in predictive reliability and substantially better diagnostic utility. Therefore, the practical question is not whether TuneAhead is the absolute cheapest method, but whether this **small extra cost** is justified by the clear improvement in screening quality; our results suggest that it is.
>
> We will revise the final version to explicitly separate the negligible **one-time offline predictor fitting cost** from the **per-run deployment-time screening cost** that matters in practice.
>
> **(2) Hyperparameter adjustment and effort when switching models/datasets.**
> TuneAhead does **not** rely on heavy hyperparameter tuning. For meta-dataset construction, hyperparameters are treated as part of the input: we define a modest predefined LoRA hyperparameter space and include each \((D_i, H_j)\) configuration as a candidate run, rather than first tuning to a dataset-specific optimum. For predictor training, the LightGBM hyperparameters are **fixed across all experiments**, e.g., `learning_rate=0.05` and `num_leaves=4`.
>
> So the direct answer for “Does switching to another model or dataset require significant effort in parameter tuning?” is: switching to another **model or dataset** does **not** require significant effort in parameter tuning on the predictor side; we reuse the same feature pipeline and the same LightGBM configuration. We will revise the final version to make this clearer.
>
> **(3) Could TuneAhead be useful for efficient data selection in LLM fine-tuning or instruction tuning?**
> **Yes**—at the **dataset/run level**, we believe TuneAhead is already useful for efficient data selection and instruction tuning. One intended use of the framework is to rank candidate dataset–hyperparameter settings before committing the full fine-tuning budget.
>
> More importantly, our paper already includes a direct empirical example of this utility. In the failure case study, TuneAhead’s SHAP analysis identified **high duplication** and **high outlier ratio** as major contributors to a poor predicted run. We then applied **semantic de-duplication** and **outlier removal**, and the final MMLU score improved from **20.2% to 48.7%**. This shows that TuneAhead is not only useful for screening candidate runs, but can also provide actionable signals for **dataset refinement** in instruction tuning.
>
> We agree that the current paper does not yet implement **instance-level** selection, and we will make this scope clearer in the final version. We will also strengthen the discussion of TuneAhead as a practical tool for **coarse-grained data selection** and **data refinement**.
>
> Again, we thank the reviewer for the thoughtful suggestions. We believe these clarifications will make the paper more precise and practically useful, and we will incorporate them in the final version.

---

> > ### Author Rebuttal · Reviewer_wy6n · 2026-04-03
> >
> > Thank you for the detailed response. I believe the paper's findings will be of great interest to the LLM research community, and I will maintain my original positive score.

---

> > > ### Author Response · Authors · 2026-04-07
> > >
> > > Thank you very much for your kind follow-up and for maintaining your positive score. We greatly appreciate your careful reading, encouraging assessment, and constructive feedback. Your comments have been very helpful in improving the clarity and practical positioning of the paper.

---

### Decision · Program_Chairs · 2026-04-30

**Decision:**

Accept (regular)

**Comment:**

This paper proposes TuneAhead, a framework for predicting LLM fine-tuning performance prior to full training by combining static dataset descriptors with dynamic features from a short probe run. The method utilizes a gradient-boosting predictor and provides interpretability via SHAP attributions.

The reviewers acknowledge the practical significance of the problem and the strong empirical evaluation across 1,300+ runs, where the method outperforms existing baselines in prediction accuracy and offers actionable diagnostics for failure cases. However, concerns were raised regarding the generalizability across model families, the empirical nature of the fixed probe length, potential data leakage in feature selection, and the clarity of compute-saving claims.

During the rebuttal, the authors clarified that feature selection was strictly confined to training/validation data, addressing the leakage concern. They also provided new preliminary results demonstrating that lightweight calibration enables effective intra-family transfer, mitigating generalization worries. The authors committed to refining the manuscript to explicitly scope limitations regarding cross-family transfer, the fixed probe heuristic, and the threshold-dependency of compute savings. While one Reviewer maintained a weak reject stance citing remaining scope limitations, the other three reviewers found the responses satisfactory, with two upgrading their scores to "Accept."

Given the solid experimental evidence, the practical utility of the proposed screening tool, and the authors' willingness to clarify the method's boundaries in the final version, the consensus supports acceptance. The paper makes a valuable contribution to resource-efficient LLM adaptation, conditioning that the final revision accurately reflects the discussed limitations.